# Analysis of Non-Polar Low-Molecular Metabolites in Citron (*Citrus medica* L.) Peel Essential Oil at Different Developmental Stages and a Combined Study of Transcriptomics Revealed Genes Related to the Synthesis Regulation of the Monoterpenoid Compound Nerol

**DOI:** 10.3390/ijms26189034

**Published:** 2025-09-17

**Authors:** Jie Luo, Wanting Zhong, Yayi Xiong, Jian Wang

**Affiliations:** 1College of Traditional Chinese Medicine, Chongqing Medical University, Chongqing 400016, China; 2023121966@stu.cqmu.edu.cn (J.L.); 2024122021@stu.cqmu.edu.cn (Y.X.); 2Department of Chinese Materia Medica, Chongqing University of Chinese Medicine, Chongqing 402760, China; 3Chongqing Key Laboratory of Chinese Medicine for Prevention and Cure of Metabolic Disease, Chongqing Medical University, Chongqing 400016, China

**Keywords:** citron, terpenoids, developmental stages, volatile compounds, biosynthesis, transcriptomics, gene expression

## Abstract

Citron (*Citrus medica* L.) peel essential oil is rich in terpenoids, but their composition and formation mechanisms remain unclear. We analyzed volatile compounds and gene expression in citron peel across three developmental stages using GC-MS and RNA-seq. GC-MS identified 91 volatiles, with γ-Terpinene and D-Limonene the most abundant; terpenes dominated (95.60–87.72%) but decreased with maturation, with 19 differential terpenoid metabolites. Differential expression analysis identified 10,238 genes with significant changes across three stages, including 256 TPS genes and 656 transcription factors. Via WGCNA and correlation analysis, 10 gene–metabolite pairs were identified, including 5 key genes and 2 transcription factors linked to nerol synthesis. These findings provide a framework for future research on citron genetics, terpenoid identification, and regulatory mechanisms of terpenoid biosynthesis.

## 1. Introduction

Citrus species, belonging to the Rutaceae family, are among the most widely cultivated fruit crops globally, distributed across the monsoon regions from western Pakistan to northern China, etc. [1]. *Citrus medica* L., regarded as one of the oldest ancestors of the Citrus genus, is believed to have originated in southeastern China, primarily found in regions such as Fujian, Yunnan, etc. [2]. According to the Chinese Pharmacopoeia, the medicinal sources of Citron primarily include *Citrus wilsonii* Tanaka and *Citrus medica* L., with the latter being more commonly used [3]. China is rich in Citron germplasm resources, exhibiting considerable intraspecific variation and significant differences in fruit characteristics. At least 11 types of Citron have been identified [4]. Variations in traits and chemical composition among different Citron types pose challenges for their identification and standardized medicinal application.

Citron has high edible, processing, ornamental [5] and medicinal values, widely used in traditional Chinese medicine and folk therapy. Its mature dried fruit is the medicinal part, used for treating vomiting, phlegm and cough, etc. Volatile oil refers to oil-like liquids extracted from natural plants that contain monoterpenes, sesquiterpenes, their derivatives, and small molecular aliphatic compounds such as hydrocarbons, alcohols, and esters. The rich aroma of Citron is attributed to the accumulation of these components, with monoterpenes and sesquiterpenes contributing significantly to the characteristic fragrance. In the traditional Chinese medicine system, volatile oils have various pharmacological effects such as analgesia, bacteriostasis and anti-inflammation. The volatile oil components of Citron peel are complex, predominantly consisting of terpenes such as limonene, β-pinene,γ-terpinene, linalool [6], etc. These terpenoid volatile compounds have been demonstrated to possess antioxidant [7,8], antibacterial [9], anti-inflammatory [10], antitumor [11], hypolipidemic [12], and hepatoprotective activities [13], making them important bioactive constituents in essential oils. Additionally, terpenes like thymol [14], linalool [15], limonene [16], β-pinene [17], citral [18], and geraniol [19] have been reported to exhibit antidiabetic effects in diabetic rats, including lowering blood sugar levels and restoring plasma insulin levels.

Terpenoid compounds in essential oils are vital sources of natural bioactive substances, yet their biocontrol mechanisms remain largely unknown. Terpenoids, as secondary metabolites [20], are derived from basic C5 isoprenoid units, namely isopentenyl diphosphate (IPP) and dimethylallyl diphosphate (DMAPP). These precursors are synthesized through two main pathways in plant cells: the mevalonate pathway (MVA) in the cytoplasm and the methylerythritol phosphate pathway (MEP) in chloroplasts [21]. IPP and DMAPP undergo head-to-tail condensation to form various intermediates, including geranyl diphosphate (GPP), farnesyl diphosphate (FPP), and geranylgeranyl diphosphate (GGPP) [22]. Terpene synthases (TPS) play a crucial catalytic role in these pathways, producing diverse terpenoid products from GPP, FPP, and GGPP. Radioactive labeling experiments have confirmed that monoterpenes such as nerol and geraniol are reported to be synthesized via the mevalonate pathway [23]. The TPS gene family exhibits high diversity and complex catalytic mechanisms, enabling the synthesis of a wide range of terpenoids [24]. TPS genes are classified into seven main subfamilies, including TPS-a, TPS-b, TPS-c, TPS-d, TPS-e/f, TPS-g, and TPS-h, with TPS-a and TPS-d associated with sesquiterpene synthases and TPS-b linked to monoterpene synthases [25]. Recent studies have advanced our understanding of TPS genes; for instance, the gene BsTPS2 in Boswellia serrata has been shown to catalyze the conversion of GPP and FPP to (S)-(+)-linalool [26]. Additionally, the terpene synthase EoTPS has been demonstrated to convert geranyl diphosphate to geraniol in vitro [27].

Transcription factors (TFs) also play important regulatory roles in the biosynthesis of terpenoids, with various TF families, such as WRKY, bHLH, MYB, bZIP, and AP2/ERF, involved in this process [28]. For example, TFs CpMYC2 and CpbHLH13 may regulate the synthesis of monoterpene (linalool) and sesquiterpene (β-caryophyllene) in *Chimonanthus praecox* [29]. In *Jasminum sambac*, high expression levels of the WRKY family TF JsWRKY51 have been linked to the accumulation of the secondary metabolite β-ocimene, indicating its regulatory role in aromatic substance synthesis [30]. Research by Li et al. [31] on the regulation of volatile compounds in *Citrus sinensis* Osbeck revealed that the TPS CitTPS16 catalyzes E-geraniol in vitro and is regulated by the AP2/ERF family TF CitERF71. Similarly, the biosynthesis of the sesquiterpene (+)-valencene is catalyzed by CsTPS1 and positively regulated by CitAP2.10 from the AP2/ERF family [32]. Currently, the regulatory mechanisms of terpenoid biosynthesis in citrus plants remain underexplored, highlighting the importance of future research in this area for scientific value and potential applications.

Numerous studies have conducted transcriptomic analyses of various medicinal plants, including *Citrus medica* var. *sarcodactylis* [33], *Panax quinquefolius* L. [34], *Panax notoginseng* (Burkill) F.H. Chen ex C.H. Chow) [35], Polygonatum cyrtonema Hua [36], and Coptis chinensis Franch [37]. Xu et al. [38] conducted an in-depth analysis of the transcription levels of TFs genes during the development of *Citrus medica* var. *sarcodactylis* using RNA-Seq technology, they identified 13 TPS genes and 15 transcription factors associated with the formation of volatile terpenoids, establishing a foundational framework for future studies on the regulatory networks of terpenoid compounds in this species. Given the current lack of transcriptomic data for Citron, this study employed the Illumina HiSeq 4000 platform to perform de novo transcriptome sequencing at three developmental stages of the fruit, creating a transcriptome database. This work provides essential baseline information for future investigations into the biosynthesis and regulatory mechanisms of terpenoid active components in Citrus essential oils.

The experimental samples in this study were collected from Midu County, Dali, Yunnan, China, where a local specialty citron named “Siji” is produced. Currently, there is a lack of research on the essential oil chemical composition of this Citrus variety. Therefore, it is crucial to conduct relevant studies to enrich the genetic resources of Citron and promote its rational and standardized utilization.

This study aims to reveal the dynamic accumulation patterns of terpenes and their essential oil components during three developmental stages of Citron: green fruit (QF), yellow-green fruit (QHF), and ripe fruit (YF). We will establish a comprehensive chemical profile of volatile compounds to provide scientific evidence for determining optimal harvest times. By integrating transcriptomic techniques, we will analyze differentially expressed genes closely related to the accumulation of terpenoid active components, identifying key terpenoid synthase (TPS) genes and transcription factors (TFs). This study will lay a foundation for future research on the regulatory network of citron terpenoid biosynthesis and terpenoid metabolic engineering.

## 2. Results

### 2.1. Volatile Compound Analysis of Citron Peel Essential Oil at Different Developmental Stages

To elucidate the dynamic changes in secondary metabolites in citron fruit peel essential oil across three developmental stages, GC-MS analysis was conducted on the essential oils from these stages (Figure 1a). Detailed information on the volatile compounds in the peel essential oils identified at each stage and the total ion current chromatograms are included in the Appendix A, see Appendix A, respectively. A total of 91 volatile compounds were identified during the entire developmental phase (Appendix A). The primary classes of volatile compounds included 15 monoterpenes, 12 sesquiterpenes, 15 oxygenated monoterpenes, 5 oxygenated sesquiterpenes, and 44 other compounds such as fatty acids, aldehydes, esters, and ketones (Appendix A). Among these 91 volatile compounds, 38 were detected in all three stages, 1 compound was unique to the QF stage, and 43 were identified in the YF stage (Figure 1b). The most abundant volatile compound was γ-Terpinene (12.91–20.30%), followed by D-Limonene (8.68–12.05%), (+)-3-Carene (7.35–11.63%), Citral (7.07–8.61%), Neral (5.68–6.96%), Terpinolene (4.78–6.03%), β-pinene (2.24–2.88%), β-Myrcene (2.29–2.68%), and α-pinene (1.17–1.72%). Terpenoids were the predominant compound class in citron essential oil, comprising 87.72–95.60% of the total oil content, with monoterpenes accounting for 55.88–67.23%, sesquiterpenes for 3.31–4.73%, oxygenated monoterpenes for 23.45–32.38%, and oxygenated sesquiterpenes for only 0.16–0.93%. As development progressed, the percentage of terpenoids in citron fruit peel essential oil gradually decreased, with 95.60% in the QF stage, 94.48% in the QHF stage, and 87.72% in the YF stage. In particular, the content of monoterpenoid components showed the most significant changes; during the fruit ripening stage, a large number of esters, fatty acids, and other components appeared. (Figure 1d). From the QF to YF stages, the number of terpenoids, aldehydes, and esters showed an increasing trend, with 48 terpenoids and their derivatives, 11 aldehydes, and 12 esters detected in the YF stage. Additionally, 6 ketones, 6 fatty acids, and 3 alcohols were also identified during this period (Appendix A).

To clearly illustrate the differences in volatile compounds in citron fruit peel essential oil across the three stages, a heatmap was constructed to present an overview of the volatile compounds detected at different developmental stages (Figure 2). Hierarchical clustering primarily divided the data into two clusters. The QF and QHF stages exhibited similarities in the variety and distribution of volatile compounds, thus they were grouped into one large cluster. The YF stage had the highest number of volatile components, with significant changes in key constituents such as γ-Terpinene, D-Limonene, and (+)-3-Carene compared to QF, justifying its classification as a separate cluster. The average percentages of γ-Terpinene in QF and YF were 20.30% and 12.86%, showing a decrease of 7.44%; D-Limonene exhibited decreases from 12.04% to 8.53% (3.53% reduction); (+)-3-Carene showed a decline from 11.63% to 7.23% (4.4% reduction). Terpenoid compounds, including γ-Terpinene, D-Limonene, (+)-3-Carene, Terpinolene (6.03%), and β-Bisabolene (1.53%), were most abundant in QF, highlighted as distinct red regions in the heatmap. In the column direction, volatile components such as (−)-cis-beta-Elemene to 5-Hepten-2-one and 6-methyl were also delineated as notable red rectangular regions. This area primarily comprised 42 volatile compounds, including esters, aldehydes, fatty acids, and ketones, which were exclusively detected in YF and correlated with high concentrations during this stage. In YF essential oil, the proportions of esters, aldehydes, fatty acids, and ketones were 2.63%, 1.88%, 2.76%, and 0.78%, respectively, compared to 1.60%, 0.11%, 0.18%, and 0.13% in QF oil. The quantities of these volatile compounds in YF increased by 10, 10, 5, and 5, respectively (Appendix A).

### 2.2. Identification of Key Differential Volatile Metabolites in the Peel Essential Oil of Citron at Different Developmental Stages

The significant changes in volatile compounds observed in Figure 2 prompted the identification of differential metabolites during various developmental stages. We conducted pairwise comparisons of the volatile components in citron peel essential oil, identifying 19 key terpenoid differential metabolites (Figure 3). Compared to the yellow fruit (YF), the yellow-green fruit (QHF) showed significant reductions in 12 terpenes, including nerol, geraniol, and caryophyllene, with log2 fold changes (log2FC) less than −1 and adjusted *p*-values below 0.01. Notably, nerol, geraniol, and caryophyllene decreased by 4.16, 2.8, and 2.64 times, respectively (Appendix A). The comparison between green fruit (QF) and YF revealed 18 terpenoid differential metabolites, with an additional 7 compounds, including α-phellandrene and D-limonene, significantly downregulated. The most substantial reductions were observed for γ-terpinene, (+)-3-carene, and α-phellandrene, decreasing by 2.73, 2.72, and 2.67 times, respectively. Other compounds showed reductions ranging from 2.0 to 2.40 times. Additionally, two terpenoid derivatives, citronellyl acetate and geranyl acetate, decreased, while n-hexadecanoic acid levels increased (Table 1 and Appendix A). No significant differential metabolites were observed between QF and QHF (Appendix A). In PCA, PC1 explained 73.3% of data variance (R^2^X = 0.733) and PC2 20.3% (R^2^X = 0.203). The PCA score plot showed distinct separation of citron peel essential oil samples across three developmental stages, indicating significant inter-group differences. The variable loading plot revealed some volatile components clustered and close to samples, consistent with differential analysis results (Appendix A).

To clearly illustrate the changes in 19 key secondary terpenoid metabolites across QF, QHF, and YF stages, we constructed and visualized a heatmap (Figure 3). In brief, the hierarchical clustering analysis revealed a prominent red block in QF, indicating a larger area of accumulation compared to YF. The QHF stage also exhibited two distinct red areas compared to YF. All 19 key terpenoid differential metabolites were found to be at higher concentrations in QF and QHF than in YF, showing significant downregulation from QF to YF, consistent with the results of differential analysis.

### 2.3. Transcriptome Sequencing and Analysis

#### 2.3.1. Transcriptome Sequencing Data Analysis

This study utilized the Illumina sequencing platform to perform unassembled transcriptome sequencing on citron peel samples at different developmental stages. After filtering out low-quality reads, high-quality reads were obtained. Sequencing quality statistics for the nine samples are shown in Table 2. A total of 402,040,024 high-quality reads were generated, yielding 59.41 Gb of effective data. The sequencing data indicated that Q20 quality scores exceeded 96.95%, Q30 quality scores exceeded 90.89%, and GC content ranged from 44.95% to 46.71%. These results demonstrate that high-quality reads were produced, providing a reliable foundation for subsequent transcriptome assembly and analysis, thereby ensuring the accuracy and validity of future research.

To obtain more accurate expression levels and transcript counts, a unique identifier code (UID) was introduced during the library construction phase to trace the origin of individual molecules and reduce data bias. Each molecule was randomly tagged with a unique UID to ensure accurate differentiation of distinct original molecules in subsequent analyses. After sequencing, an internal UID processing software was employed to classify and merge reads with similar sequences, aiming to correct errors and eliminate duplicates generated by PCR amplification, thus obtaining more precise molecular sequences and expression data. high-quality reads underwent UID deduplication and quality control, followed by rRNA filtering and removal of overrepresented sequences and error correction to ensure high data quality. Finally, TRINITY software was utilized to assemble the remaining high-quality reads into transcripts, yielding high-quality transcriptome sequences. Detailed statistics on UID deduplication and the high-quality data are presented in Table 3. Ultimately, after a series of quality control steps, a total of 200,644,160 high-quality, non-duplicate reads were obtained for subsequent transcript assembly.

In this study, a total of 101,796 transcript sequences were assembled, with an average length of 1320.1 bp. The continuity metrics N50 and N90 of the assembly were 2137 bp and 594 bp, respectively. The average GC content of these transcripts was 40.4%. To reduce redundancy and enhance data representativeness, clustering based on sequence similarity was performed, selecting the longest transcript from each cluster as a representative. Ultimately, 49,596 non-redundant unigenes were obtained. The average length of the unigenes was 872.5 bp, with N50 and N90 values of 1591 bp and 331 bp, respectively, and a GC content of 40.4%. Detailed statistical data on all transcript assemblies are provided in Table 4.

To evaluate the completeness and assembly quality of the de novo transcriptome of *Citrus medica* L. (citron), we performed BUSCO (Benchmarking Universal Single-Copy Orthologs) analysis using the embryophyta_odb10 dataset, which contains 1614 conserved single-copy orthologous gene groups specific to embryophytes. The results showed that 79.8% of the BUSCO groups were identified as complete (C), including 78.7% as complete and single-copy (S) and 1.1% as complete and duplicated (D) (Appendix A). Specifically, 1288 complete BUSCOs were detected, among which 1270 were single-copy and only 18 were duplicated, indicating minimal redundancy in the transcriptome assembly. Additionally, 11.2% of the BUSCOs were classified as fragmented (F), corresponding to 181 gene groups, while 9.0% (145 gene groups) were missing (M). These metrics demonstrate that the assembled transcriptome captures a high proportion of conserved embryophyte genes with low redundancy, providing a reliable foundation for subsequent functional annotation and gene expression analysis.

#### 2.3.2. Alignment and Annotation Information of Unigene Sequences

During the transcriptome assembly, we employed the BLAST sequence alignment tool for homology searches across multiple databases to obtain corresponding annotation information. Results indicated that the Rfam database provided the most annotations, covering 95,979 genes, while the NR database annotated only 25,044 genes. Additionally, the Uniprot, Pfam, and eggNog databases annotated 80,800, 75,201, and 72,146 genes, respectively. Functional annotations were also performed for the transcripts using KEGG and GO databases. In total, 100,765 genes received annotations, providing a crucial foundation for subsequent functional analyses (Table 5).

To investigate the functional roles of annotated transcripts, particularly their potential involvement in the biosynthesis of secondary metabolites, we conducted GO and KEGG pathway enrichment analyses. In the GO database, a total of 57,106 genes were annotated, while the KEGG database annotated 22,160 genes. The GO annotations were categorized into three main classes: Biological Process (BP), Cellular Component (CC), and Molecular Function (MF). Specifically, the BP category included 30,227 genes with 1926 GO terms; the CC category comprised 33,634 genes with 589 GO terms; and the MF category involved 71,217 genes with 1413 GO terms (Appendix A). The most enriched entries in these categories were: “ATP binding” in MF with 9022 genes; “integral component of membrane” in CC with 15,512 genes; and “regulation of transcription, DNA-templated” in BP with 1668 genes. Figure 4 illustrates the top five significantly enriched entries and their corresponding gene numbers across the BP, CC, and MF categories.

In the KEGG pathway annotation analysis, the related pathways identified from gene annotations were classified into six categories: Organismal Systems, Metabolism, Environmental Information Processing, Genetic Information Processing, Human Diseases, and Cellular Processes. The Environmental Information Processing category encompassed 7136 genes across 39 pathways; Cellular Processes included 7274 genes within 31 pathways; Genetic Information Processing covered 8302 genes with 23 pathways; Organismal Systems comprised 10,207 genes across 87 pathways; and Human Diseases involved 13,698 genes, also covering 87 pathways. Notably, the Metabolism category exhibited the highest enrichment, involving 36,658 genes and 148 related pathways (Appendix A). The top five enriched pathways within the Metabolism category were “Metabolic pathways,” “Biosynthesis of secondary metabolites,” “Biosynthesis of antibiotics,” “Microbial metabolism in diverse environments,” and “Carbon metabolism,” with gene counts of 8260, 3943, 1927, 1541, and 1181, respectively.

Figure 5 presents the top five significantly enriched pathways within the six categories, along with their corresponding gene counts. Notably, we focused on five pathways closely related to terpenoid biosynthesis: “Terpenoid backbone biosynthesis,” “Ubiquinone and other terpenoid-quinone biosynthesis,” “Monoterpenoid biosynthesis,” “Sesquiterpenoid and triterpenoid biosynthesis,” and “Diterpenoid biosynthesis,” in addition to “Limonene and pinene degradation.” These pathways involve 205, 203, 27, 46, 38, and 35 genes, respectively, with associated terpenoid synthesis proteins numbering 31, 22, 4, 6, 9, and 2. These findings provide valuable insights into the potential roles of transcripts in secondary metabolism and terpenoid synthesis.

### 2.4. Transcription Factor Annotation

To delineate the profile of plant transcription factor (TF) families in the citrus peel, we annotated all unigene sequences against the PlantTFDB database, using *Citrus sinensis* as the reference species. Sequence alignments with TF protein sequences from PlantTFDB led to the identification of 57 TF families, comprising a total of 1645 TFs (Appendix A). The bHLH family was the most abundant, with 135 members; followed by the ERF, C2H2, MYB, NAC, and MYB-related families, which contained 122, 112, 91, 89, and 84 TFs, respectively. Families with relatively few members included HB-PHD, S1Fa-like, Whirly, and NZZ/SPL, with counts of 3, 2, 2, and 1, respectively. Figure 6 illustrates the distribution of the 57 TF families in the citrus peel transcriptome, reflecting their abundance and diversity, thereby providing a foundation for understanding the transcriptional regulatory network.

### 2.5. Analysis of Differentially Expressed Genes in Citrus Fruit Peel at Different Developmental Stages

To elucidate the differential expression of transcripts across various developmental stages, we conducted pairwise comparisons of gene expression for three developmental stages (QF vs. QHF, QHF vs. YF, QF vs. YF) using the R package DESeq2. Before conducting differential analysis, PCA was first performed to verify the uniformity among biological replicates in the transcriptomic dataset. Biological replicate samples from the same period were clearly clustered together, indicating the reliability of the experimental data. The first two principal components collectively explained 69.4% of the total variance (Appendix A). In the comparison between QF and QHF, after filtering low-expression genes, 17,026 unigene non-zero FPKM values were included, resulting in 3541 downregulated DEGs and 4266 upregulated DEGs. For the QHF vs. YF comparison, 16,983 unigene non-zero FPKM values were analyzed, yielding 3836 downregulated DEGs and 3032 upregulated DEGs. In the QF vs. YF comparison, 16,879 unigene non-zero FPKM values were considered, leading to 1648 downregulated DEGs and 1966 upregulated DEGs. A total of 10,238 DEGs were identified across the three stages (Appendix A). Figure 7 panels a–c illustrate the differential expression profiles of genes for the three comparisons, highlighting significant expression differences across stages. Panels d–f present volcano plots showing the relationship between expression level changes (log2 Fold Change) and statistical significance (−log10 *p*-value) for differential genes, with all DEGs exhibiting |log2 Fold Change| greater than 1 and *p*-values less than 0.05.

To reveal the expression dynamics of terpene synthase (TPS) genes across three developmental stages, we identified 256 TPS genes with significant differential expression at the QF, QH, and YF stages. These genes encompass key enzymes involved in various terpene biosynthetic pathways, including monoterpenes, sesquiterpenes, and diterpenes, as well as genes related to the degradation of terpenes like limonene and pinene. Additionally, a total of 656 transcription factors (TFs) exhibited significant differential expression across the stages, with representative families including WRKY, bHLH, ERF, and MYB (Appendix A). These findings highlight the complexity of terpene biosynthesis and its regulatory networks during the development of citrus fruits. 

### 2.6. Weighted Gene Co-Expression Network Analysis (WGCNA) of DEGs

In our analysis of differentially expressed genes (DEGs) from fruit peel samples across different stages, we identified a total of 10,238 DEGs, which exhibited distinct expression patterns at different time points. Similar expression patterns across time points indicate their coordinated roles in specific biological processes or functions. Indeed, the synthesis of particular secondary metabolites in plants is commonly regulated by multiple genes collectively. The biosynthetic pathways for certain metabolites often involve a series of enzymatic reactions, with each enzyme corresponding to one or more coding genes; the expression levels of these genes collectively influence product formation. Transcription factors also regulate the expression of key synthetic genes, contributing to the complex regulatory network of plant secondary metabolites. To further investigate the roles of these DEGs across different stages, we conducted a weighted gene co-expression network analysis (WGCNA) to identify gene modules with similar expression patterns among the 10,238 DEGs, thereby uncovering potentially functionally related gene sets and providing new insights into the regulatory network of terpene synthesis. Through a series of matrix constructions and the approximation of scale-free network construction, we identified five gene modules from dynamic hierarchical clustering (Figure 8), which were distinguished by color. The largest module, the turquoise module, contains 4528 genes; followed by the blue module with 4490 genes; and the brown and yellow modules with 741 and 477 genes, respectively. The grey module is the smallest, comprising only 2 genes.

To assess the biological significance of the different color modules, we calculated and visualized the Pearson correlation coefficients between module characteristic genes and external sample traits (specifically, the concentrations of 19 key differential terpenoid metabolites in this study). The five color modules exhibited high correlations with the 19 key terpenoid metabolites, with most modules showing correlation coefficients |corr| > 0.6. We also calculated the *p*-values for these correlations, all of which were less than 0.01. As shown in Figure 9, the yellow module displayed a negative correlation with most key metabolites, while the brown module also exhibited negative correlations with many of them. The blue module demonstrated a strong positive correlation with monoterpenes such as 3-Carene, and the turquoise module showed a very strong positive correlation with nerol and geraniol, with correlation coefficients |corr| > 0.9. To further elucidate the reasons behind the strong correlations between modules and metabolites, as well as the biological significance of each module, we re-annotated the genes contained in the different color modules using KEGG and focused on the enrichment results and biological relevance of these genes in terpenoid metabolic pathways.

After KEGG annotation, the turquoise module was associated with 105 gene records related to terpenoid biosynthesis pathways, including Terpenoid backbone biosynthesis, Ubiquinone and other terpenoid-quinone biosynthesis, Sesquiterpenoid and triterpenoid biosynthesis, Diterpenoid biosynthesis, Monoterpenoid biosynthesis, and Limonene and pinene degradation. The blue module was linked to 121 gene records related to terpenoid biosynthesis, while the yellow and brown modules were annotated with 22 and 8 records, respectively. The grey module did not yield any annotations. In total, 228 genes were annotated, accounting for 41.16% of the total annotated terpenoid biosynthesis genes (554). Specifically, 114 terpenoid backbone synthesis genes were annotated, representing 55.60% of the total (205); 76 terpenoid quinone synthesis genes accounted for 37.44% of the total (203); 12 sesquiterpenoid and triterpenoid synthesis genes comprised 26.09% of the total (46); 17 diterpenoid synthesis genes represented 44.74% of the total (38); and 15 monoterpenoid synthesis genes constituted 42.86% of the total (35). Additionally, 22 genes related to limonene and pinene degradation accounted for 81.48% of the total (27). The genes and detailed annotation records for the turquoise and blue modules are summarized in Appendix A. The distinct biological functional separation observed in the annotation information of different modules greatly intrigued us and will be further discussed in Section 3.

### 2.7. Correlation Analysis of Differentially Expressed Genes and Terpenoid Metabolites at Different Developmental Stages

The significant enrichment of sesquiterpene and monoterpene synthesis genes in the blue and turquoise modules, along with the co-expression relationships among genes within these modules, is likely to provide insights into the regulatory relationships governing terpenoid metabolite biosynthesis. This study explores potential biosynthetic relationships between differentially expressed genes and differential metabolites using Spearman correlation analysis. The analysis was conducted using the built-in R function cor.test to calculate the Spearman correlation coefficients and their corresponding significance levels (*p*-values). To control the false positive risk associated with multiple testing, we applied FDR correction to the *p*-values, ensuring the reliability of the correlation results. A total of 10 pairs of significantly correlated gene–metabolite pairs were identified, with selection criteria set at an absolute correlation coefficient |corr| > 0.6 and a corrected *p*-value (FDR) < 0.05. Among these relationships, the correlation coefficients ranged from 0.97 to 1, with all FDR values below 0.05, indicating a very strong correlation (Appendix A). Figure 7 illustrates the correlation between differentially expressed genes and key differential terpenoid metabolites among these 10 pairs, providing important clues for elucidating the potential regulatory mechanisms of terpenoid biosynthesis.

The correlation results between genes in the blue and turquoise modules and terpenoid differential metabolites are shown in Figure 10, where we identified 10 significantly correlated gene–metabolite pairs. Notably, 9 of these genes were significantly associated with nerol, with 5 genes (marked in blue) showing a significant negative correlation (|corr| > 0.97) and 4 genes exhibiting a significant positive correlation (|corr| > 0.97). Other compounds only displayed a correlation with one gene. Among the 9 genes significantly related to nerol, TRINITY_DN20691_c0_g1, identified as a TPS gene, is involved in the biosynthesis of terpenoid backbones; TRINITY_DN11666_c0_g1, highly similar to the *Citrus sinensis* gene sequence orange1.1g030419m, is classified as a FAR1 family transcription factor; and TRINITY_DN24161_c0_g1, closely resembling orange1.1g044013m, is classified as a C2H2 family transcription factor. The blue and red colored genes exhibit a clear opposite trend in their expression patterns. For ease of discussion, we have designated TRINITY_DN11666_c0_g1 as CmFAR1-tf and TRINITY_DN24161_c0_g1 as CmC2H2-tf.

To explore the regulatory relationship between significantly correlated genes and nerol, we depicted a line graph showing the dynamic changes in these genes at three developmental stages (Figure 11), with expression levels represented by FPKM values. The y-axis values of the points denote the average expression or content level of the gene or metabolite at each stage. In Figure 11, three genes, including TRINITY_DN11666_c0_g1, demonstrate an opposite trend compared to nerol across the three stages, while four genes, such as TRINITY_DN24806_c0_g1, show a similar trend to nerol. TRINITY_DN20233_c0_g1 exhibits an upward trend from the QH to YF stage, but the changes are not significant, with average FPKM values of 6.66 and 7.45 for QH and YF, respectively. Notably, from QF to QH, the expression level of TRINITY_DN20233_c0_g1 significantly decreased by approximately 2.41-fold. TRINITY_DN19320_c0_g1 and TRINITY_DN19318_c1_g8 appear to show strong negative correlation with nerol in Figure 11 and calculations, but they are likely to be strongly positively correlated, possibly due to the small sample size leading to unstable correlation estimates. From QF to QH, TRINITY_DN19320_c0_g1 increased by about 3.92-fold, and TRINITY_DN19318_c1_g8 increased by 16.33-fold. From QH to YF, TRINITY_DN19320_c0_g1 decreased by approximately 35.23-fold, and TRINITY_DN19318_c1_g8 decreased by 16.33-fold. Information regarding the sequences and expression levels of the seven significantly correlated genes is provided in Appendix A.

In summary, the dynamic changes in these seven genes are consistent with nerol content and align with the results of the correlation analysis. TRINITY_DN11666_c0_g1, TRINITY_DN20691_c0_g1, and TRINITY_DN19318_c1_g4 exhibit strong negative correlations with nerol, while TRINITY_DN24806_c0_g1, TRINITY_DN20234_c5_g1, TRINITY_DN24161_c0_g1, and TRINITY_DN20233_c0_g1 show strong positive correlations. These results suggest that nerol expression may be influenced by the expression of the five genes in Figure 11 and is also regulated by two transcription factors.

## 3. Discussion

Citrus plants, as an ancient plant species, are characterized by high bud mutation rates, absence of apomixis, and common natural hybridization, along with a long cultivation history and wide distribution. The taxonomy and evolutionary history of Citrus are highly complex due to prolonged evolution. Currently, citron (*Citrus medica* L.) is regarded, based on phylogenetic analyses, as one of the three ancestral species of the Citrus genus [39]. It is widely believed that Citrus plants originated in the tropical and subtropical regions of Southeast Asia and subsequently spread to other continents [40]. In China, citron naturally occurs in the southwestern and southeastern areas and has been extensively cultivated and domesticated across the country, resulting in a diverse array of medicinal citron germplasm, their volatile chemical components show significant variations. In the fruits of the citron from Motuo, Tibet, D-limonene has been reported as the predominant volatile compound, with a relative content of 53.00%, followed by 3-carene (8.21%), nerol (5.10%), and geraniol (4.92%) [41]. In the essential oil of ripe citron (*Citrus wilsonii* Tanaka) cultivated in Jingjiang, Jiangsu Province, the relative content of D-limonene is 50.53%, followed by p-cymene (16.40%), γ-terpinene (8.70%), β-ocimene (5.03%), and β-pinene (3.35%) [42]. In contrast, the essential oil of Finger citron (*Citrus medica* L. var. *sarcodactylis*) from Jinhua, Zhejiang Province, comprises Limonene (45.36%), γ-terpinene (21.23%), dodecanoic acid (7.52%), and β-bisabolene (3.23%) [43]. In the essential oil extracted from the mature peel of citron (*Citrus medica* L. var. *macrocarpa* Risso) from southern Fars Province, Iran, the content of limonene reaches 78.86%, with neral (3.69%), geranial (4.92%), linalool (1.79%), and geraniol (1.76%) as secondary components [44]. The samples used in this study were collected from citron cultivated in Midu County, Dali, Yunnan, where GC-MS analysis of the mature peel essential oil revealed major components including γ-terpinene (12.86%), D-limonene (8.53%), (+)-3-carene (7.23%), o-cymene (6.64%), citral (8.46%), neral (6.46%), terpinolene (4.67%), β-pinene (2.88%), and terpinen-4-ol (2.27%). The composition of Dali Midu citron essential oil differs significantly from that of citrons in other regions, with the difference in D-limonene content being particularly notable. The γ-terpinene content in Dali Midu citron is higher than in the first three but lower than that in Jinhua. Oxygen-containing monoterpenes like citral and neral are present in higher concentrations in Dali Midu citron than in the other four varieties, where they are either low or absent. Additionally, the number of volatile compounds detected varies significantly, with 51, 44, 28, and 28 compounds identified in citrons from Motuo, Jingjiang, Fars Province, and Jinhua, respectively, compared to 91 compounds in Dali Midu citron essential oil. This difference may be attributed to geographical environments or genetic backgrounds. Comprehensive investigations are needed to better understand the chemical diversity and potential value of citron, providing a scientific basis for resource conservation, rational use, and applications in the medicinal and fragrance industries.

This study reveals a patterned variation in secondary metabolite content in the essential oils of citron peels across different developmental stages. From the QF to YF stages, the percentage of terpenes in the essential oils gradually decreases, with average terpene percentages of 95.60%, 94.48%, and 87.72% for QF, QHF, and YF stages, respectively. The reduction from QF to QHF is minor (1.12%), showing no significant change, while a notable decline of 7.88% occurs from QHF to YF. Monoterpenes dominate the terpene composition, accounting for 67.23%, 57.98%, and 55.88% in QF, QHF, and YF stages, respectively. There is a reduction of 9.34% from QF to QHF and a further decrease of 2.1% from QHF to YF, primarily driving the overall decline in terpene percentage. The quantity of monoterpenes remains stable across developmental stages, while sesquiterpenes show an increase in number (by 5) from QF to YF, despite a decrease in total content. Oxygenated sesquiterpenes also increase in both quantity (by 5) and percentage (by 0.77). As fruits mature from QHF to YF, more aldehydes and esters emerge in the essential oils, with the number of aldehydes, ketones, and fatty acids increasing by 10, 5, and 5, respectively, and their percentages rising by 1.70%, 0.67%, and 2.43% (Appendix A). Notably, the volatile content, including Limonene, declines, with percentages of 12.04%, 9.20%, and 8.53% in QF, QHF, and YF stages, respectively. Similarly, γ-terpinene shows a peak in QF (20.30%) and drops to its lowest in YF (12.86%). Xu et al. [38] observed a decrease in the content of volatile compounds, primarily Limonene, during the maturation of Buddha’s hand (*Citrus medica* L. var. *sarcodactylis*). Ghani et al. [8] also noted a significant reduction in Limonene levels and monoterpene content as citron developed, aligning with our findings. Wu et al. [44] reported a decrease in Limonene alongside an accumulation of ketones during fruit maturation. In their study on Satsuma mandarin (*Citrus unshiu* Marcov cv *Owari*), Gao et al. [45] found a significant decline in total free volatile concentrations over 40 days of fruit maturation, with sesquiterpenes decreasing and some volatiles, like hexanal, emerging only in later stages. This dynamic change parallels our results, where sesquiterpene content decreases and certain volatiles are absent in QF and QHF but appear in YF, including various aldehydes and esters. Although factors such as collection time, geographical climate, genetic variation, and biotic stress may influence the dynamic changes in volatile secondary metabolites across different developmental stages, further research is warranted to provide comprehensive insights into the dynamic chemical profiles of plant volatile secondary metabolites.

Study on differentially expressed genes in different developmental stages, we performed weighted gene co-expression network analysis (WGCNA), which revealed four main modules with similar expression patterns. After re-annotation, we observed distinct characteristics within these modules. In the turquoise module, we found significant enrichment of 43 terpene backbone synthesis genes, 42 terpene quinone biosynthesis genes, as well as 11 sesquiterpene and triterpene biosynthesis genes, 7 diterpene biosynthesis genes, and additional genes related to monoterpene synthesis and the degradation pathways of limonene and pinene. The blue module showed enrichment of 60 terpene backbone synthesis genes, 30 terpene quinone biosynthesis genes, along with 14 monoterpene biosynthesis genes and 17 limonene and pinene degradation genes. The brown module was significantly enriched with 10 diterpene biosynthesis genes alongside 4 terpene backbone synthesis genes and 3 terpene quinone biosynthesis genes, as well as 1 sesquiterpene biosynthesis gene and 4 related to limonene and pinene degradation. The yellow module contained only 7 terpene backbone synthesis genes and 1 terpene quinone biosynthesis gene. Interestingly, the yellow module exhibited a strong negative correlation with 19 key terpene metabolites, likely due to the absence of annotated terpene biosynthesis genes, suggesting that other highly differentially expressed genes may influence these metabolites. Additionally, the blue module was notably enriched with 17 genes linked to the degradation pathways of limonene and pinene, accounting for 62.96% of the total annotated genes in this pathway (27). A strong correlation (−0.78) was observed between the blue module and D-limonene. Notably, genes TRINITY_DN12579_c0_g1 and TRINITY_DN20057_c1_g2 were significantly upregulated (approximately 27-fold and 2.83-fold, respectively) from QHF to YF, both involved in the limonene and pinene degradation pathway (ko00903), primarily relating to the enzyme aldehyde dehydrogenase (NAD+) [EC:1.2.1.3] (k00128). This aligns with observations of significant reductions in D-limonene content in citrus essential oils during maturation. Furthermore, the blue module, enriched with 14 monoterpene synthesis genes, showed strong positive correlations with other monoterpenes, such as (+)-3-Carene and Terpinolene, and weak correlations with sesquiterpenes like (1Z,4E)-germacrene B. In contrast, the brown module, enriched with 10 diterpene synthesis genes and lacking monoterpene genes, showed strong negative correlations with nine monoterpenes and a negative correlation with D-limonene (−0.63). The turquoise module exhibited weak positive correlations with sesquiterpenes (correlation coefficients 0.39–0.52) and a strong positive correlation with a monoterpene gene, with coefficients ranging from 0.66 to 0.98. The grey module presented a strong positive correlation with most key terpene metabolites, despite limited annotations and low gene counts, highlighting its potential significance for future research due to minimal influence from other highly differentially expressed genes (average correlation coefficient 0.64). Overall, this study successfully identified gene sets with certain degrees of distinct characteristics and similar expression patterns through WGCNA, which may provide directional guidance for subsequent research. It should be noted that the WGCNA in this study relied solely on gene expression data from 9 biological replicates. This limited dataset may compromise the reliability and generalizability of results, failing to fully reflect the true co-regulatory relationships of gene expression across developmental stages. Expanding biological replicates and including samples from more developmental time points will better capture dynamic changes in gene expression and metabolite accumulation, enhance the robustness of co-expression networks and accuracy of module function annotations. This will further facilitate exploring dynamic regulatory mechanisms of key genes in terpene synthesis pathways, clarifying genuine intra-module gene co-expression relationships and spatiotemporal matching between gene expression changes and terpene accumulation. Ultimately, it provides a more reliable molecular network basis for elucidating the functional roles of TPS and TFs in terpene biosynthesis.

In the correlation analysis of differentially expressed genes and terpene metabolites, seven genes were found to be significantly correlated with nerol, exhibiting consistent dynamic changes in expression with nerol levels across the QF, QH, and YF stages. Among these genes, we identified one terpene synthase (TPS) gene and two transcription factors from different families. TRINITY_DN20691_c0_g1, a terpene backbone synthesis gene, is primarily involved in the biosynthesis of terpene precursors via the MEP (ko00900) as an isoprene synthase (k12742), which catalyzes the conversion of IPP and DMAPP into isoprene, serving as a key enzyme linking primary (MEP) and secondary terpene metabolism. From QF to QH, the expression level of TRINITY_DN20691_c0_g1 decreased approximately 57.66-fold, while from QH to YF, it increased by about 320.64-fold, coinciding with nerol levels increasing by approximately 1.59-fold and then decreasing by 4.16-fold. The synthesis of monoterpenes requires GPPS (GPP synthase) to convert DMAPP and IPP into GPP, the precursor for monoterpenes. The high expression of TRINITY_DN20691_c0_g1 may competitively deplete IPP/DMAPP, leading to reduced monoterpene synthesis sources and establishing an inverse regulatory relationship with nerol. Notably, we did not observe significant expression increases in TPS genes associated with other upstream terpene backbone biosynthesis genes that could enhance IPP/DMAPP sources. Among the identified transcription factors, CmFAR1-tf from the FAR1 family downregulated by 5.89-fold from QF to QH and upregulated by 7.35-fold from QH to YF, exhibited a negative regulatory effect on nerol. Conversely, CmC2H2-tf from the C2H2 family was upregulated by 32.15-fold from QF to QH and downregulated by 30.76-fold from QH to YF, demonstrating a positive regulatory effect on nerol. These two transcription factors, CmFAR1-tf and CmC2H2-tf, likely interact with two to three other genes to jointly regulate nerol biosynthesis across different developmental stages.

The regulatory mechanisms of FAR1 and C2H2 family transcription factors on terpene and other secondary metabolites remain unclear. However, existing studies indicate that these transcription factors are involved in plant growth, development, and the biosynthesis and accumulation of various metabolites. For example, Ma et al. demonstrated that FAR1 can transcriptionally activate the starch-debranching enzyme gene ISA2, positively regulating starch synthesis [46]. Additionally, FAR1 has been shown to positively regulate the biosynthesis of myoinositol, thereby alleviating oxidative stress damage [47]. The C2H2 family transcription factor GmZFP7 has been reported to regulate the biosynthesis of isoflavones in soybean (*Glycine max*), promoting isoflavone accumulation through the high expression of isoflavone synthase 2 (GmIFS2) and flavanone 3 β-hydroxylase 1 (GmF3H1) [48]. Liu et al. investigated the C2H2 transcription factor AtGIS, which regulates glandular trichome development in *Nicotiana tabacum* (*Xanthi*) through GA signaling. The high expression of AtGIS significantly increased the yield of glandular trichomes in tobacco leaves, thereby enhancing the accumulation of volatile compounds such as nicotine and cembratriene-4,6-diol [49]. In addition to the discussed FAR1, differential expression of C2H2 family transcription factors was observed across the three developmental stages. Other transcription factor families, such as WRKY, ERF, MYB, bHLH, and AP2, also exhibited significant expression changes during these periods. Although no transcription factors were identified that perfectly matched the accumulation patterns of key terpene compounds, they have been reported to participate in the regulation of plant secondary metabolite synthesis. The bHLH family is one of the largest transcription factor families in plants, with approximately 28,698 bHLH transcription factors identified in the PlantTFDB [50]. Previous studies have reported the role of bHLH transcription factors in regulating terpene biosynthetic pathways. For instance, Shang et al. successfully identified two bHLH transcription factors, B1 (bitter leaf) and Bt (bitter fruit), which regulate the synthesis of the monoterpene compound cucurbitacin [51]. Additionally, the transcription factor bHLH3 is involved in the biosynthesis of triterpenoid saponins in *Glycyrrhiza uralensis* [52]. MYB transcription factors have also been found to play roles in the transcriptional regulation of terpenoid compounds. For example, Wu et al. discovered that AaMYB1 positively regulates the accumulation of artemisinin, a sesquiterpene lactone, while the expression of another transcription factor, AaMYB15, decreases artemisinin content [53]. Additionally, PnMYB2, isolated from *Panax notoginseng*, is believed to regulate the biosynthesis of triterpenoid ginsenosides [54]. The AP2/ERF family is primarily involved in plant stress responses and the regulation of secondary metabolites. WRKY transcription factors represent one of the most prominent transcription factor families in plants. Research on WRKY proteins has spanned three decades, and these factors are known to participate in various signaling pathways, including plant hormone signaling, mitogen-activated protein kinase (MAPK) cascades, reactive oxygen species (ROS) signaling, and the biosynthesis of secondary metabolites [55]. Gao et al. reported that the WRKY transcription factor LcWRKY17 enhances the transcriptional activity of the terpene synthase gene LcTPS42 in *Osmanthus fragrans* by binding to the W-box in its promoter, promoting the biosynthesis of monoterpenes [56]. Additionally, PkdWRKY and PksWRKY regulate the biosynthesis of picroside, with PkdWRKY positively regulating and PksWRKY negatively regulating the key regulatory genes PkHMGR and PkDXS in this pathway [57]. The regulatory network governing the biosynthesis of plant terpenoid metabolites is complex, potentially influenced by TPS and various transcription factors. Future research strategies are needed to provide new insights into this intricate regulation.

The essential oil from citron fruits is rich in various terpenoid compounds, showcasing potential as natural bioactive molecules. However, significant variations in chemical composition exist due to differences in geographic environment, climate, and genetic resources. Therefore, further foundational research is necessary for their rational application. In this study, we identified genes and regulatory transcription factors related to the biosynthesis of the terpenoid metabolite nerol, providing a basis for understanding its molecular mechanisms. In the future, we will characterize these 10 gene–metabolite pairs, with a focus on TPS genes showing significant strong correlation with nerol. These genes will be cloned and expressed in *E. coli* to produce recombinant enzymes for in vitro enzymatic activity assays. By using GPP/FPP as substrates, we will determine their catalytic product profiles. Additionally, in vivo catalytic activity will be evaluated through expression in engineered yeast strains, aiming to clarify the division of labor of these genes in terpenoid metabolic pathways and explain the dynamic changes and biosynthetic regulatory mechanisms of monoterpenes and sesquiterpenes during *Citrus medica* development. For example, Jin et al. [58] proved that PnCPS is a caryophyllene synthase by cloning PnTPS, and Anand et al. [59] confirmed that TPS contig 19414 (TPS1) functions as a cis-β-terpineol synthase in terpenoid biosynthesis.Using the DNA-binding domains of CmFAR1-tf and CmC2H2-tf as baits, we will screen the *Citrus medica* cDNA library to identify their binding TPS gene promoters. Direct binding activity between transcription factors and target gene promoters will be validated in vitro via EMSA, such as whether CmFAR1-tf activates the nerol synthase gene expression, thereby revealing the molecular mechanisms of transcriptional regulatory hierarchy in terpenoid metabolism and providing new targets for artificial regulation. This work is also expected to enable high-efficiency terpenoid production based on synthetic biology, breaking through the low-yield bottleneck of plant-derived terpenoids and providing green production technology for natural bioactive compounds.

With the increasing demand for valuable natural products from plants, significant efforts have been made to enhance the yield of secondary metabolites. The advent of molecular biology has deepened our understanding of the molecular basis of secondary metabolite biosynthesis and the regulatory mechanisms plants employ in these biochemical processes. Based on this knowledge, two additional strategies—synthetic biology-based synthesis and breeding integrated with molecular biology—offer new possibilities for effectively promoting the production of these important natural products. Research on the genetic regulatory relationships governing the synthesis of natural products is an effective approach to increasing the content of plant metabolites [60]. Future studies revealing the variations in plant secondary metabolite synthesis will further facilitate these strategies and accelerate the utilization of valuable natural product resources from plants.

## 4. Materials and Methods

### 4.1. Plant Materials

The experimental *Citrus medica* L. samples were collected from local cultivation bases in Jili Village, Jili Town, Midu County, Dali Bai Autonomous Prefecture, Yunnan Province, China (25°14′29.43″ N, 100°34′07.47″ E). According to the morphological characteristics of *Citrus medica* L. recorded in reference [4] and the data from the Plant Species Information Platform of the Institute of Botany, Chinese Academy of Sciences (http://www.iplant.cn/, accessed on 15 July 2024), the plant was identified as *Citrus medica* L. by the authors and Professor Jian Wang from Chongqing University of Chinese Medicine. Fruits were sampled at three developmental stages: green fruit (QF), yellow-green fruit (QHF), and yellow fruit (YF), with three biological replicates per stage. Each biological replicate was derived from three different plants, with six fruits of similar height, size, and color harvested from each plant to form one replicate sample. In total, nine biological replicates were obtained across the three stages. QF samples were collected in April, QHF in August, and YF in December. Upon harvesting, fruits were immediately rinsed with tap water, air-dried, washed three times with DEPC water (G8010, Adamas life, Shanghai, China), and further air-dried. For each stage, three fruits were randomly selected from the three biological replicates, and approximately 1 g of mid-peel tissue was cut into 0.2 cm × 1 cm pieces. These were placed in DNase-free EP tubes, flash-frozen in liquid nitrogen for 3 min, and stored at −80 °C for subsequent total RNA extraction. Three biological replicates per stage were used for transcriptomic sequencing. Remaining fruits were batch-processed in biological replicates for essential oil extraction from the peel. The fresh *Citrus medica* L. pericarp samples were cut into 1 cm-long × 2 mm-thick small pieces, then subjected to oven drying at 50 °C for 10 h with intermittent ventilation. This resulted in three batches of pericarp samples for essential oil extraction, with each batch used for detection at each developmental stage. Each batch of samples was tested in triplicate.

### 4.2. Extraction and Collection of Fruit Peel Essential Oil

The essential oil extraction from Citron peel was performed using steam distillation with a Soxhlet extractor. Peel samples from the three biological replicates at each developmental stage were used for extraction. Approximately 200 g of peel was placed in a round-bottom flask, with pure water as the solvent, ensuring the solvent level did not exceed two-thirds of the container’s volume. The extraction solvent is n-hexane (14153M, Adamas, Shanghai, China), with a liquid-to-material ratio of 10:1 (mL/g). The temperature is set at 65 °C, and the number of cycles per hour is 4. Each extraction lasted for a minimum of 5 h. The essential oils obtained from each biological replicate were dehydrated using anhydrous Na_2_SO_4_ (82667L, Adamas, Shanghai, China), and stored at 4 °C for future use.

### 4.3. GC-MS Analysis of the Chemical Components of Citron Peel Essential Oil at Different Developmental Stages

Before conducting GC-MS analysis of essential oils, fifty microliters of essential oil was placed in a 2ml sample vial, to which 1ml of n-hexane was added. The mixture was shaken thoroughly and set aside for use.

GC-MS analysis was conducted using an Agilent 7890B system (Santa Clara, CA, USA). The chromatographic column was an HP-5 ms Ultra Inert (30 m × 0.25 mm × 0.25 μm). A sample volume of 1 μL was injected with a column flow rate of 1 mL/min. The injector temperature was set at 200 °C, and the initial column temperature was maintained at 60 °C for 3 min, then increased to 250 °C at a rate of 4 °C/min, holding for 10 min. The split ratio was 50:1. The electron impact (EI) ion source operated at 70 eV, with the MS quadrupole temperature at 150 °C and the ion source temperature at 230 °C. The scanning range was 15–500 *m*/*z*, with a scan speed of 5.7 scans/s. The solvent delay was set to 3 min, and high-purity helium was used as the carrier gas.

Volatile compound identification was performed by matching the electron collision mass spectra of individual compounds with those in the NIST 17.0 mass spectral library (https://chemdata.nist.gov/, accessed on 1 September 2024). The Kovats retention index for each volatile compound was calculated and compared with those reported by other researchers. Using the same instrument conditions, diluted C7–C33 n-alkane standard solutions (zhongce biaowu#BWY0821002, Zhongce Biaowu, Chengdu, Sichuan, China) were injected into the GC-MS for Kovats retention index calculation based on carbon number and retention time. The percentage content of each volatile compound was quantified using peak area analysis.

### 4.4. Transcriptome Sequencing

#### 4.4.1. RNA Extraction

Total RNA from citron peel samples was extracted using the TRIzol reagent (Invitrogen#15596018CN, Invitrogen, Carlsbad, CA, USA). First, 200 mg of the peel sample was rapidly ground into a powder in liquid nitrogen and transferred to a centrifuge tube. An appropriate volume of TRIzol was added, and the mixture was allowed to stand at room temperature for 5 min. The sample was then centrifuged at 12,000 rpm for 5 min, and the precipitate was discarded. Chloroform (75915R, Adamas, Shanghai, China) was added at a ratio of 200 μL per mL of TRIzol, mixed thoroughly, and left at room temperature for 15 min. The mixture was centrifuged at 12,000× *g* for 15 min at 4 °C, and the upper aqueous phase was transferred to a new centrifuge tube. Isopropanol (75885W, Adamas, Shanghai, China) was added at a ratio of 0.5 mL per mL of TRIzol, mixed, and allowed to stand at room temperature for 5–10 min. The sample was then centrifuged at 12,000× *g* for 10 min at 4 °C, discarding the supernatant to observe RNA pelleting at the bottom of the tube. Next, 1 mL of 75% ethanol per mL of TRIzol was added, and the tube was gently mixed to resuspend the pellet. The sample was centrifuged at 8000× *g* for 5 min at 4 °C, and the supernatant was carefully discarded. After air-drying or vacuum-drying for 5–10 min at room temperature, an appropriate volume of TE buffer (Takara#T9121, Takara, Kusatsu-shi, Japan) was added to dissolve the RNA sample. RNA concentration was quantified using a spectrophotometer (NanoDrop^TM^ One^C^, Thermo Fisher, Shanghai, China), and RNA quality was assessed to ensure it met library preparation requirements.

#### 4.4.2. cDNA Library Construction and Sequencing

RNA samples passing quality checks will be used to construct cDNA libraries for transcriptome sequencing. Approximately 5 μg of total RNA is used to enrich and isolate mRNA using magnetic beads with Oligo (dT). The isolated mRNA is then fragmented using Fragmentation Buffer. The resulting mRNA fragments serve as templates for synthesizing single-stranded cDNA with six-base random primers. Subsequently, a buffer, dNTPs (#KGF2202-500, KeyGEN, Nanjing, Jiangshu, China), and DNA polymerase I (18010017, Thermo Fisher, Shanghai, China) are added to synthesize double-stranded cDNA. The purified double-stranded cDNA undergoes end repair, A-tailing, and adapter ligation, which includes UID adapters at the 5′ end of the cDNA. Magnetic beads are employed to recover fragments of the desired size, followed by PCR amplification and purification of the PCR products to prepare the cDNA library. Quality control of the constructed library is performed using agarose gel electrophoresis with the following parameters: gel concentration at 1%, voltage at 170 V, electrophoresis time of 20 min, and sample volume of 4 μL. The library is quantified using Qubit™ 2.0 (Q33327, Thermo Fisher, Shanghai, China) to determine its concentration. Upon passing quality control, the library will be sequenced on an Illumina sequencer, targeting an output of 6G of data based on effective concentration.

#### 4.4.3. Sequence Data Processing and Transcriptome Assembly

The raw image data obtained from Illumina HiSeq (San Diego, CA, USA) sequencing is converted into sequence data (FASTQ format) through base calling, resulting in the initial sequencing data files, referred to as raw reads. Quality control of the sequencing data is essential after sequencing. Fastp (version 0.23.0) [@fastp] software is utilized to filter out low-quality reads, sequences with high N rates, and those that are too short, yielding high-quality reads. The high-quality reads undergo further processing, including UID deduplication, followed by filtering for rRNA using software such as SortMeRNA(version 4.3.7), Rcorrector(version 1.0.7), and Fastuniq(version 1.1), as well as error correction and deduplication. The final set of high-quality reads is assembled de novo using the Trinity software (version 2.15.0) through three steps: “cocoon, pupa, and butterfly,” resulting in the final transcript sequences. The longest transcript is designated as unigene for subsequent downstream transcriptomic analyses.

### 4.5. Functional Annotation of Unigenes and Annotation of Plant Transcription Factors

Using various databases, including UniProt (https://www.uniprot.org), the Non-Redundant Protein Sequence Database (NR, https://www.ncbi.nlm.nih.gov/protein, accessed on 10 December 2024), the Protein Family Database (Pfam, http://pfam.xfam.org, accessed on 10 December 2024), the RNA Families Database (Rfam, http://rfam.xfam.org/, accessed on 10 December 2024), and the Evolutionary Genealogy of Genes (eggNOG, http://eggnogdb.embl.de, accessed on 10 December 2024), we conducted homologous searches of known sequencing data using the BLAST tool (version 2.16.0) for sequence similarity comparisons of unigenes. Additionally, we performed Gene Ontology (GO) classification (http://geneontology.org/, accessed on 26 December 2024) and KEGG metabolic pathway analysis (Kyoto Encyclopedia of Genes and Genomes, https://www.kegg.jp/, accessed on 26 December 2024) to obtain functional annotation information for the unigenes. Transcription factors (TFs) were annotated using the Plant Transcription Factor Database (PlantTFDB, https://planttfdb.gao-lab.org/, accessed on 15 May 2025), with *Citrus sinensis* selected as the reference species.

### 4.6. Integrated Analysis of Transcriptome and Metabolome

Differentially expressed genes (DEGs) were identified using R (version 4.3.2) and the DESeq2 package (version 1.40.2), comparing transcriptome sequencing data between the QF and QHF stages, as well as between the QHF and YF stages. The selection criteria were set to an adjusted *p* value (*p*.adj) of less than 0.05 and |log2 fold change| greater than 1, indicating a fold change (FC) greater than 2. Gene expression levels at each stage were quantified using FPKM (Fragments Per Kilobase of transcript per Million mapped reads). Differentially expressed metabolites were analyzed using R (version 4.3.2) and the limma package (version 3.56.2), comparing GC-MS data of volatile compounds in fruit peel between the QF and QHF stages, as well as between the QHF and YF stages. The selection threshold was similarly set to *p*.adj < 0.05 and |log2 fold change| > 1. Weighted Gene Co-expression Network Analysis (WGCNA, version 1.73) was performed to identify gene sets with similar expression patterns. Finally, Spearman correlation analysis was conducted on the identified DEGs and differential metabolites to explore potential relationships between them.

### 4.7. Statistical Analysis

Mean and standard deviation were calculated using Microsoft Excel (version 3.8.0.12160). All visualizations, including heatmaps, volcano plots, MA plots, and correlation plots, were generated and combined using R (version 4.3.2). Differential analysis was conducted using the DESeq2 package (version 1.40.2), with data first checked for NA and missing values. Missing values were filled with the mean, and all data underwent normalization prior to differential analysis. To reduce noise, low-expression genes with total expression levels below 10 were filtered out. Differential analysis of volatile metabolites was performed using the limma package (version 3.56.2), where continuous data were converted into a format suitable for linear modeling. A design matrix was created, and Bayesian estimation was applied to the linear model results, controlling the false discovery rate (FDR) to manage false positive rates. Calculations for the t-statistic and Bayesian factor (B) were performed. Log2 fold change (log2FC) for all differentially expressed genes and metabolites was computed, along with significance testing metrics *p*-value and adjusted *p*-value (*p*.adj), with the latter corrected using the Benjamini–Hochberg method. The standard error of log fold change (lfcSE) and the statistical test statistic (stat) for hypothesis testing were also calculated. Correlation analysis was conducted using the corrplot package (version 0.95) and the built-in R function cor.test. PCA analysis was performed using the statistical software SIMCA (version 14.1). The completeness and quality of the assembled transcriptome were evaluated using BUSCO (version 5.5.0) with the embryophyta_odb10 dataset as the reference.

## 5. Conclusions

Our study on the dynamic profiling of chemical components and transcriptomics of citron peel essential oil at different developmental stages provides new insights into the biosynthesis and regulation of terpenoids. During citron fruit development, the percentage of terpenes in the essential oil gradually decreases, while the content of monoterpenes declines, and the quantity of sesquiterpenes increases. At maturity, there is a significant emergence of compounds such as aldehydes and esters. A total of 91 volatile compounds were identified by GC-MS, with γ-Terpinene, D-Limonene, (+)-3-Carene, o-Cymene, Citral, Neral, and Terpinolene being the major constituents of citron peel essential oil. Using RNA-seq technology, we constructed a non-reference transcriptome database for citron peel samples, identifying 1645 transcription factors (TFs) from 57 families. Differential analysis revealed 19 key terpenoid volatile components that vary during different developmental stages. Seventy-five TPS genes and 656 TFs exhibited significant expression across three periods, uncovering the complex dynamic regulatory mechanisms of terpenoid secondary metabolites. Through WCGNA and Spearman correlation analysis, we identified five key genes significantly associated with the accumulation patterns of the monoterpene nerol, including two transcription factors from the FAR1 and C2H2 families, CmFAR1-tf and CmC2H2-tf, which show opposing regulatory relationships. In summary, this study provides insights into the diversity of terpenoids and dynamic profiles of secondary metabolites and genes during the development of *Citrus medica* fruits. Future research will validate TPS genes strongly associated with terpenoid secondary metabolites through prokaryotic/yeast expression, elucidate the regulatory mechanisms of transcription factors CmFAR1-tf and CmC2H2-tf on terpenoid synthase genes via EMSA, and ultimately facilitate efficient green production of terpenoids based on synthetic biology.

## Figures and Tables

**Figure 1 ijms-26-09034-f001:**
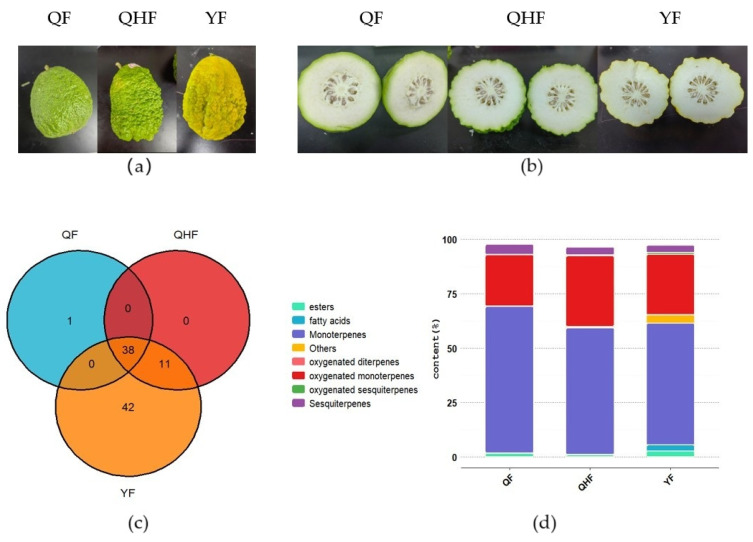
Main classes of volatile compounds in *Citrus medica* L. (**a**) Photographs of citron fruits during the QF, QHF, and YF stages. (**b**) Cross-sectional photos of Citrus reticulata fruit at three different developmental stages. (**c**) Venn diagram illustrating the volatile compounds detected in the essential oils of the peel during the QF, QHF, and YF developmental stages. (**d**) Percentage composition of major volatile compound classes in the essential oils across the QF, QHF, and YF stages, measured as the mean of three biological replicates for each stage. QF: immature fruit stage; QHF: turning fruit stage; YF: ripe fruit stage.

**Figure 2 ijms-26-09034-f002:**
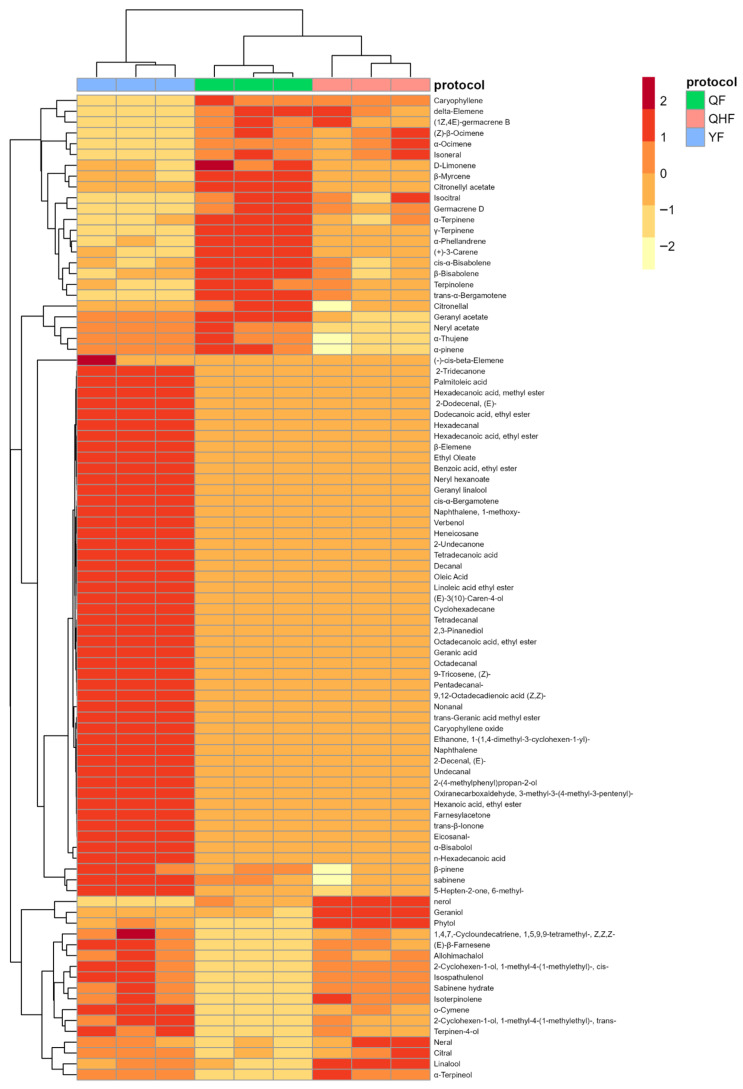
Changes in volatile compounds detected in citron across different developmental stages visualized through hierarchical clustering and a heatmap. Clustering on the left and top of the heatmap is based on Pearson correlation. The elongated legend on the right indicates the color scale (−2 to 2); red represents high concentrations, while white indicates low concentrations, with deeper colors corresponding to higher or lower amounts. The grouping protocol indicates QF, QHF, and YF for different developmental stages: QF: immature fruit stage; QHF: turning fruit stage; YF: ripe fruit stage. The hierarchical clustering and heatmap were constructed using the pheatmap package (version 1.0.12) in R.

**Figure 3 ijms-26-09034-f003:**
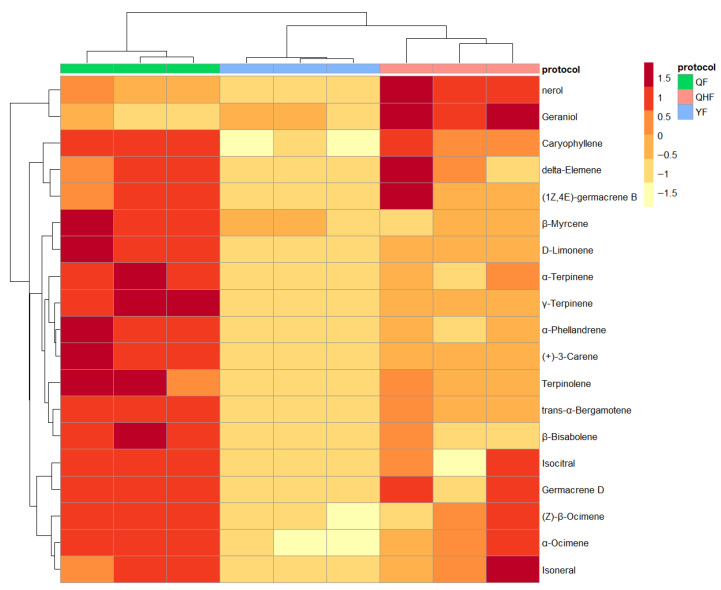
Distribution of key terpenoid differential metabolites during QF, QHF, and YF periods.

**Figure 4 ijms-26-09034-f004:**
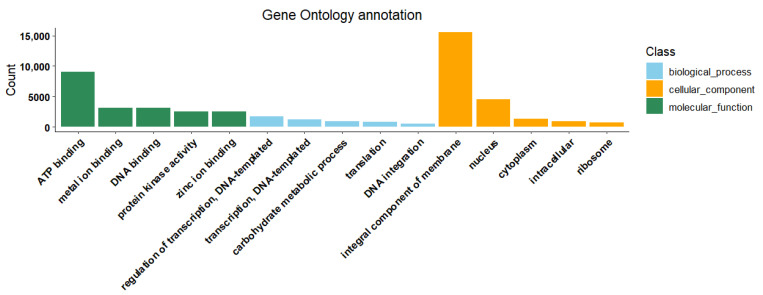
The Gene Ontology (GO) enrichment analysis results of the transcripts.

**Figure 5 ijms-26-09034-f005:**
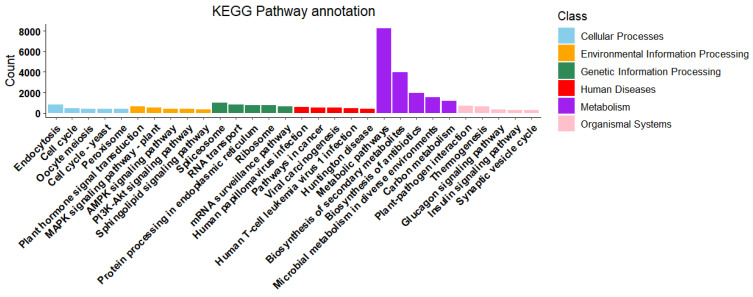
KEGG pathway annotation results of the transcripts.

**Figure 6 ijms-26-09034-f006:**
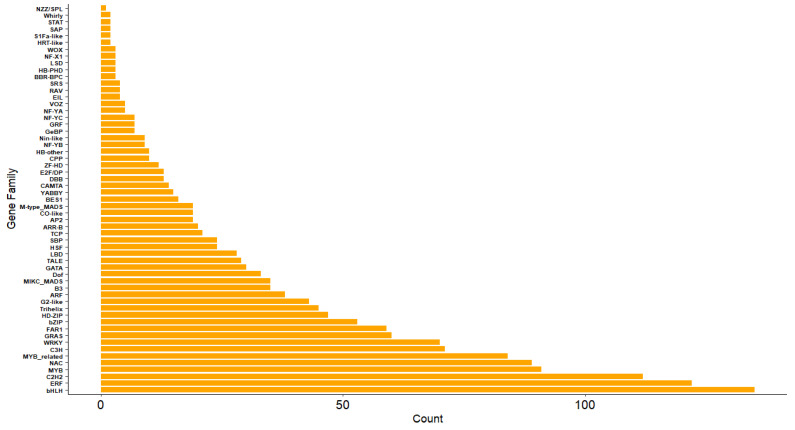
Distribution of TFs family counts.

**Figure 7 ijms-26-09034-f007:**
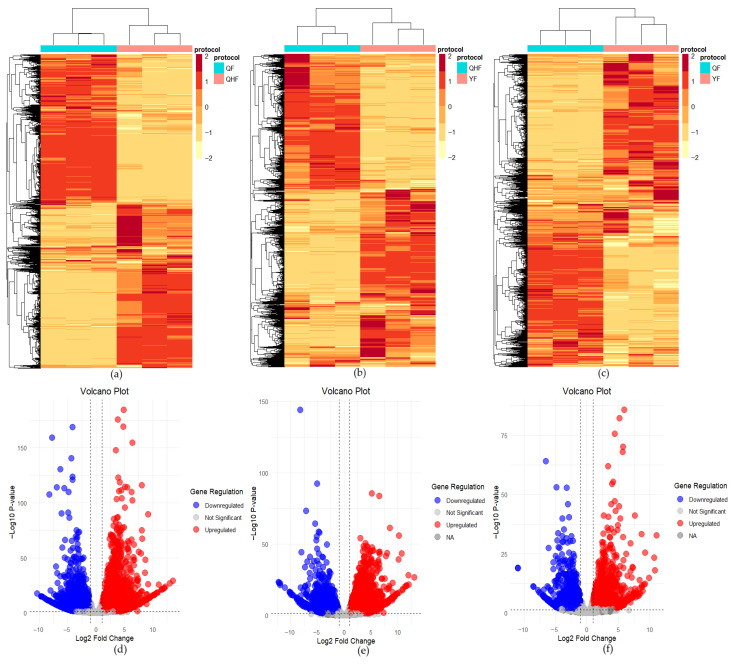
Heatmap and volcano plot of differentially expressed genes (DEGs). (**a**) Heatmap of DEGs for the QF vs. QHF comparison. (**b**) Heatmap of DEGs for the QHF vs. YF comparison. (**c**) Heatmap of DEGs for the QF vs. YF comparison. In panels A-B, the right-side color legend indicates the scale (−2 to 2), with red representing high expression levels and white indicating low expression levels; deeper colors correspond to higher or lower expression. (**d**) Volcano plot for the QF vs. QHF comparison. (**e**) Volcano plot for the QHF vs. YF comparison. (**f**) Volcano plot for the QF vs. YF comparison. In panels (**d**,**e**), red dots represent upregulated genes, blue dots indicate downregulated genes, light gray signifies genes with non-significant expression changes, and dark gray denotes genes with NA values in *p*-value calculations, indicating no expression change. The volcano plots were generated using the R package ggplot2(version 3.5.1).

**Figure 8 ijms-26-09034-f008:**
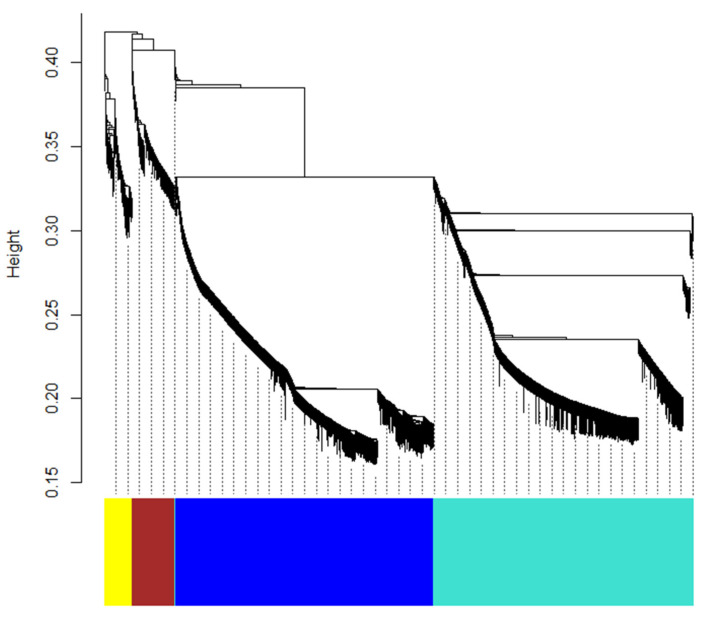
Hierarchical Clustering Dendrogram of Genes. Gene modules were identified through dynamic hierarchical tree cutting and are represented in different colors. From left to right are the yellow module, brown module, grey module, blue module, and turquoise module. Height cut = 0.25, minimal module size = 30. The weighted gene co-expression network analysis was performed using the R package WGCNA.

**Figure 9 ijms-26-09034-f009:**
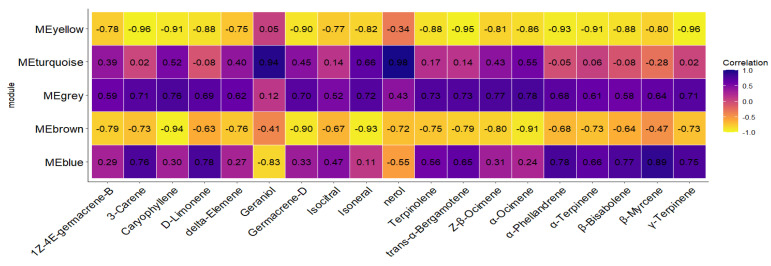
The correlation between clustered modules and differential metabolites of terpenes. The correlation coefficients between volatile compounds and module characteristic genes are presented on the right side using a color scale (ranging from −1 to 1). Dark blue and dark yellow indicate higher correlations.

**Figure 10 ijms-26-09034-f010:**
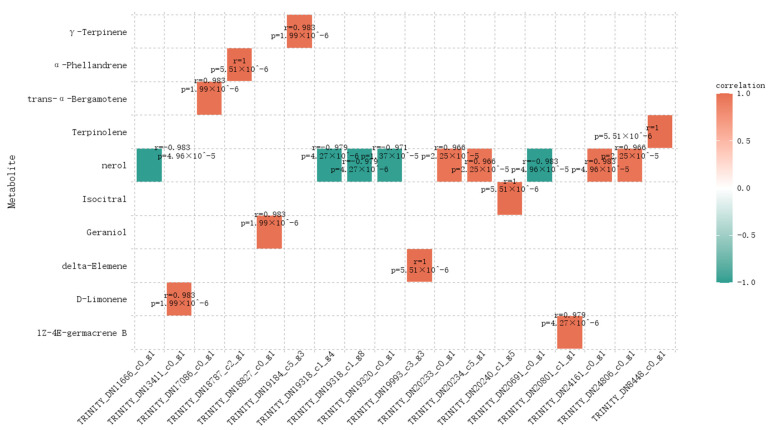
Gene–metabolite pair correlation plot. “r” represents the correlation coefficient value; “*p*” refers to the *p*-value. In the legend, the color scale is set from −1 to 1, with correlation coefficients also ranging from −1 to 1. Red indicates positive correlation, while blue indicates negative correlation, with deeper colors representing stronger correlations.

**Figure 11 ijms-26-09034-f011:**
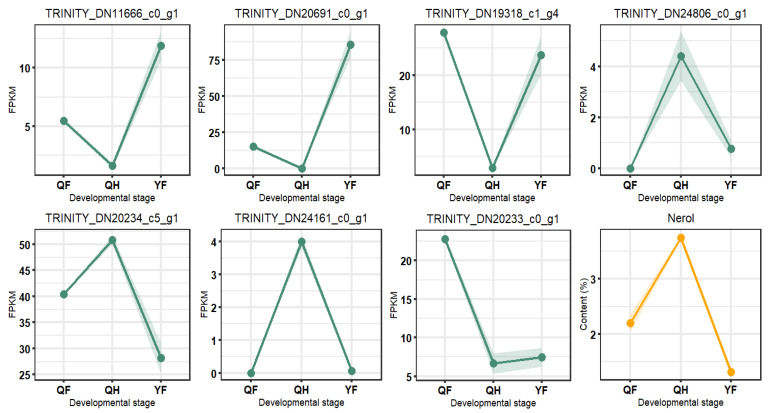
The dynamic changes in significantly correlated genes and metabolites across 3 developmental time points shown in a line plot. QF represents the green fruit stage; QH denotes the green-yellow fruit stage; and YF indicates the yellow fruit stage. In the figure, the y-values of the points are represented by their averages.

**Table 1 ijms-26-09034-t001:** Log2 fold changes (log2FC) of differential metabolites between QF and YF stages.

Compound	Log_2_FC	*p* Value	*p*.adj	B	sig
γ-Terpinene	−1.45	2.49 × 10^−43^	2.27 × 10^−41^	106.16	Down
(+)-3-Carene	−1.45	1.66 × 10^−42^	7.57 × 10^−41^	103.50	Down
α-Phellandrene	−1.42	7.29 × 10^−41^	1.66 × 10^−39^	98.25	Down
Caryophyllene	−1.42	2.42 × 10^−41^	7.33 × 10^−40^	99.77	Down
nerol	−1.39	6.45 × 10^−40^	9.78 × 10^−39^	95.25	Down
trans-α-Bergamotene	−1.35	2.06 × 10^−40^	3.75 × 10^−39^	96.80	Down
D-Limonene	−1.27	2.95 × 10^−34^	2.99 × 10^−33^	77.84	Down
Isocitral	−1.25	3.83 × 10^−39^	4.98 × 10^−38^	92.76	Down
(1Z,4E)-germacrene B	−1.19	2.67 × 10^−31^	2.43 × 10^−30^	69.07	Down
β-Bisabolene	−1.17	7.73 × 10^−31^	5.41 × 10^−30^	67.72	Down
α-Terpinene	−1.14	1.54 × 10^−29^	9.99 × 10^−29^	63.95	Down
Germacrene D	−1.13	5.97 × 10^−31^	4.53 × 10^−30^	68.01	Down
Terpinolene	−1.12	1.56 × 10^−27^	9.46 × 10^−27^	58.24	Down
n-Hexadecanoic acid	1.11	3.01 × 10^−26^	1.61 × 10^−25^	54.63	Up
delta-Elemene	−1.11	5.94 × 10^−31^	4.53 × 10^−30^	68.01	Down
α-Ocimene	−1.07	4.78 × 10^−25^	2.29 × 10^−24^	51.27	Down
Citronellyl acetate	−1.07	1.97 × 10^−38^	2.24 × 10^−37^	90.43	Down
Geranyl acetate	−1.05	2.07 × 10^−26^	1.18 × 10^−25^	55.03	Down
Isoneral	−1.01	2.78 × 10^−25^	1.40 × 10^−24^	51.88	Down
(Z)-β-Ocimene	−1.01	3.19 × 10^−23^	1.38 × 10^−22^	46.24	Down
β-Myrcene	−1.00	1.76 × 10^−22^	7.29 × 10^−22^	44.23	Down

sig: Significance of compound upregulation or downregulation; Down indicates downregulated; Up indicates upregulated. Compounds with Log2FC ≤ −1 and *p*.adj ≤ 0.05 are marked as Down, while those with Log2FC ≥ 1 and *p*.adj ≤ 0.05 are marked as Up. B refers to the Bayes Factor, which measures the reliability and significance of expression changes in compounds under different conditions; a larger value indicates greater reliability of the results. Differential analysis was conducted using the limma package in R.

**Table 2 ijms-26-09034-t002:** Transcriptome Sequencing Data Quality analysis.

Sample	R.Reads	R.Bases (G)	R.Q20 (%)	R.Q30 (%)	R.GC (%)	H.Reads	H.Bases (G)	H.Q20 (%)	H.Q30 (%)	H.GC (%)	E.Rate (%)
QF_1	40,722,928	6.15	97.24	93.29	47.1	36,309,012	5.39	98.67	95.35	45.91	89.16
QF_2	47,874,334	7.23	97.51	93.69	47	42,815,356	6.35	98.72	95.52	45.89	89.43
QF_3	39,764,314	6.00	97.13	93.09	47.13	35,436,944	5.25	98.63	95.23	45.91	89.12
QH_1	53,984,240	8.15	98.04	94.37	46.94	51,947,994	7.72	98.73	95.39	46.66	96.23
QH_2	61,495,622	9.29	98.15	94.67	47	59,437,138	8.84	98.82	95.64	46.72	96.65
QH_3	61,405,466	9.27	98.07	94.4	46.97	58,942,878	8.76	98.73	95.39	46.64	95.99
YF_1	43,738,378	6.59	95.6	89.21	45.95	40,036,546	5.83	97.56	92.39	45.55	91.54
YF_2	42,906,236	6.48	94.81	87.74	45.37	38,117,590	5.56	96.98	90.96	45.03	88.84
YF_3	43,824,350	6.62	94.76	87.7	45.3	38,996,566	5.71	96.95	90.89	44.95	88.98

R.Reads: Raw reads, the number of reads from the raw sequencing data; R.Bases: Raw Bases, the total volume of raw sequencing data; R.Q20 (%): Raw Q20 (%), the proportion of bases with quality greater than Q20 in the raw data; R.Q30 (%): Raw Q30 (%), the proportion of bases with quality greater than Q30 in the raw data; R.GC (%): Raw GC (%), the average GC content in the raw data; H.Reads: High-quality reads, the number of high-quality reads; H.Bases (G): High-quality Bases (G), the data volume of high-quality reads; H.Q20 (%): High-quality Q20 (%), the proportion of bases with quality greater than Q20 in high-quality reads; H.Q30 (%): High-quality Q30 (%), the proportion of bases with quality greater than Q30 in high-quality reads; H.GC (%): High-quality GC (%), the average GC content in high-quality reads; E.Rate (%): Effective Rate (%), the proportion of high-quality reads in the raw sequencing reads; QF_1~QF_3: Biological replicates of citron peel in the green fruit stage; QH_1~QH_3: Biological replicates of citron peel in the yellow-green fruit stage; YF_1~YF_3: Biological replicates of citron peel in the yellow fruit stage.

**Table 3 ijms-26-09034-t003:** Data Filtering Summary table.

Sample	UID Reads	Filt.rRNA Reads	Re.Overrep Reads	Correct Reads	Dedup Reads
QF_1	21,048,354	20,741,018 (98.54%)	20,702,696 (98.36%)	19,759,764 (93.88%)	16,962,776 (80.59%)
QF_2	23,886,132	23,544,430 (98.57%)	23,499,452 (98.38%)	22,445,078 (93.97%)	19,049,770 (79.75%)
QF_3	21,215,766	20,910,724 (98.56%)	20,870,194 (98.37%)	19,916,106 (93.87%)	17,075,094 (80.48%)
QH_1	31,546,066	31,403,970 (99.55%)	30,848,100 (97.79%)	29,386,708 (93.15%)	23,097,250 (73.22%)
QH_2	35,541,622	35,329,910 (99.40%)	34,324,848 (96.58%)	32,657,494 (91.89%)	25,371,004 (71.38%)
QH_3	35,171,062	35,015,082 (99.56%)	34,415,500 (97.85%)	32,806,746 (93.28%)	25,344,526 (72.06%)
YF_1	32,750,846	32,662,144 (99.73%)	32,127,036 (98.10%)	30,376,082 (92.75%)	25,751,970 (78.63%)
YF_2	29,815,338	29,756,896 (99.80%)	29,756,896 (99.80%)	28,109,788 (94.28%)	23,585,060 (79.10%)
YF_3	30,465,698	30,401,254 (99.79%)	30,401,254 (99.79%)	28,688,804 (94.17%)	24,406,710 (80.11%)

UID Reads: The number of high-quality reads retained after UID deduplication; Filt.rRNA Reads: Filtered rRNA Reads, the number of reads remaining after filtering rRNA from UID Reads, along with the proportion relative to the original UID reads; Re.Overrep Reads: Removed Overrepresented Reads, the number of reads remaining after filtering overrepresented sequences, along with the proportion relative to UID Reads; Correct Reads: The number of reads after error correction, along with the proportion relative to UID Reads; Dedup Reads: The number of deduplicated reads, along with the proportion relative to UID Reads.

**Table 4 ijms-26-09034-t004:** Transcriptome Assembly Statistics table.

Type	Trinity	Unigene
N50	2137.0	1591.0
N90	594.0	331.0
avaragelength	1320.1	872.5
maxlength	13,377.0	13,377.0
minlength	201.0	201.0
totalbase	134,384,342.0	43,271,578.0
totalcontigs	101,796.0	49,596.0
GC_content	40.4	40.4
GC_content_max	84.2	84.2
GC_content_min	7.3	7.3

Type: Trinity refers to all transcripts assembled by Trinity, while Unigene represents the longest transcripts; N50: The length of the last transcript added when all assembled transcripts are sorted from longest to shortest and cumulatively summed to reach half of the total length of the assembled transcripts; Average length: The average length of assembled transcript sequences; Max length: The maximum length of assembled transcript sequences; Min length: The minimum length of assembled transcript sequences; Total bases: The total number of bases in the assembled transcripts; Total contigs: The total number of sequences in the assembled transcripts; GC content: The average GC content of the assembled transcript sequences; GC content max: The maximum GC content among the assembled transcripts; GC content min: The minimum GC content among the assembled transcripts.

**Table 5 ijms-26-09034-t005:** Transcriptome Database Annotation Statistics table.

Data Base	Annotated Number
Uniprot	80,800
NR	25,044
Pfam	75,201
Rfam	95,979
eggNog	72,146
GO	57,106
KEGG	22,160
Total	100,765

## Data Availability

All raw data of high-throughput sequencing related to gene expression in this study have been submitted to the Sequence Read Archive (SRA) database of the National Center for Biotechnology Information (NCBI). The relevant reference information is as follows: SRA Project Accession Number (PRJNA1314466), and the data can be accessed via the link: https://www.ncbi.nlm.nih.gov/sra/?term=PRJNA1314466, accessed on 4 September 2025. This Transcriptome Shotgun Assembly project has been deposited at DDBJ/EMBL/GenBank under the accession GLIT00000000. The version described in this paper is the first version, GLIT01000000. Release date of the data: the official publication date of this study manuscript. In accordance with academic standards and data sharing principles, all datasets will be publicly available immediately after the manuscript is published.

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
