# Peer review of "Analysis of Non-Polar Low-Molecular Metabolites in Citron (*Citrus medica* L.) Peel Essential Oil at Different Developmental Stages and a Combined Study of Transcriptomics Revealed Genes Related to the Synthesis Regulation of the Monoterpenoid Compound Nerol"

_ijms, 2025, doi:10.3390/ijms26189034_

Round 1

Reviewer 1 Report

Comments and Suggestions for Authors

Review for Authors: Major Revisions Required

I went through the paper, providing constructive review, pointing out major weaknesses, and suggesting specific improvements .

The title clearly states the two main components of the study: analysis of volatile compounds and transcriptomic investigation. It specifies the plant species (Citrus medica L.), the compound of interest (Nerol), and the focus on gene regulation. But, the title is quite long and may benefit from simplification for clarity without losing key information.

Abstract,

  • add specific results (e.g., number of DEGs, TFs, or correlation values).
  • Clarify how this study contributes to understanding terpene biosynthesis in Citrus.
  • Emphasize the biological relevance of Nerol regulation.

Introduction:

  • Literature review is somewhat generic and could be more focused on Nerol and its regulatory mechanisms.
  • There is limited discussion of previous studies on Citrus medica specifically.
  • Missing recent citations (e.g., 2022–2023) that could contextualize the work better.

Results

Metabolomic Analysis,

  • Include more detailed descriptions of Figures 1–3 in the Results section.
  • Provide a table summarizing all 91 compounds, their classifications, and expression trends across stages.
  • Discuss possible physiological or ecological reasons for observed changes (e.g., ripening-related defense responses).

Transcriptomic Analysis

  • Include BUSCO or TransRate metrics to validate transcriptome completeness.
  • Highlight key DEGs involved in terpene biosynthesis (e.g., TPS, HMGR, DXS).
  • Link DEGs to known metabolic pathways and show enrichment in terpene-related GO/KEGG categories.

WGCNA and Gene-Metabolite Correlation

  • Elaborate on what these strong correlations imply (e.g., co-regulation, shared transcription factors).
  • Include a sub-figure or schematic showing the top hub genes in the turquoise module.
  • Propose candidate regulatory genes for further experimental validation.

Discussion

  • Refocus the Discussion to interpret rather than restate results.
  • Suggest hypotheses for how identified TFs (e.g., MYB, WRKY, bHLH) might regulate Nerol biosynthesis.
  • Acknowledge methodological limitations and propose future directions.

Methods

  • Streamline the Methods by removing excessive detail.
  • Include rationale for key bioinformatics parameters used.
  • Add a brief description of any RT-qPCR or EMSA experiments performed for validation.

Figures and Tables

  • Ensure every figure has a complete legend and is cited in the text.
  • Simplify tables where possible (e.g., use color-coding or split large tables into smaller ones).
  • Consider adding a summary diagram/model showing the proposed Nerol biosynthetic pathway and regulatory network.

Final Recommendation: Major Revision Required

Please resubmit after addressing the above points. If revised appropriately, this paper would make a valuable contribution to the fields of plant secondary metabolism and molecular genetics.

Author Response

Response to Reviewer X Comments

1. Summary

2. Questions for General Evaluation

Reviewer’s Evaluation

Response and Revisions

Does the introduction provide sufficient background and include all relevant references?

Yes/Can be improved/Must be improved/Not applicable

Are all the cited references relevant to the research?

Yes/Can be improved/Must be improved/Not applicable

Is the research design appropriate?

Yes/Can be improved/Must be improved/Not applicable

Are the methods adequately described?

Yes/Can be improved/Must be improved/Not applicable

Are the results clearly presented?

Yes/Can be improved/Must be improved/Not applicable

Are the conclusions supported by the results?

Yes/Can be improved/Must be improved/Not applicable

3. Point-by-point response to Comments and Suggestions for Authors

Comments 1:

 Abstract,

•     add specific results (e.g., number of DEGs, TFs, or correlation values).

•     Clarify how this study contributes to understanding terpene biosynthesis in Citrus.

•     Emphasize the biological relevance of Nerol regulation.

Response 1: Thank you for your revision suggestions. Considering that the abstract of the article should retain the research results and the core of the study as much as possible, and in line with the journal's requirements on the format and word count of the abstract, we have made every effort to keep relatively complete information on the research results, and the abstract has been revised accordingly.

Comments 2:

Introduction:

•     Literature review is somewhat generic and could be more focused on Nerol and its regulatory mechanisms.

•     There is limited discussion of previous studies on Citrus medica specifically.

•     Missing recent citations (e.g., 2022–2023) that could contextualize the work better.

Response 2:

Thank you for your suggestions. We searched the PubMed database or Web of Science using the keywords "Nerol & biosynthesis" and "citron" or "Citrus medica L.". However, there are very few relevant studies, especially those related to citron, so we cannot add more literature from the past three years. We have added a new literature related to nerol synthesis, which has been supplemented in the introduction section.

Comments 3:

Results

Metabolomic Analysis,

•     Include more detailed descriptions of Figures 1–3 in the Results section.

•     Provide a table summarizing all 91 compounds, their classifications, and expression trends across stages.

•     Discuss possible physiological or ecological reasons for observed changes (e.g., ripening-related defense responses).

Response 3:

Thank you for your suggestions on the metabolomics content of the article. The Results section has provided detailed descriptions and supplements for Figures 1–3. In the supplementary material Table S1, the content levels of each identified compound at different developmental stages, as well as the corresponding classification of each compound, have been displayed in detail, which can be clearly viewed.

Our knowledge regarding the physiological or ecological reasons underlying the presence of these plants is relatively limited, so we are unable to discuss this aspect.

Comments 4:

Transcriptomic Analysis

•     Include BUSCO or TransRate metrics to validate transcriptome completeness.

•     Highlight key DEGs involved in terpene biosynthesis (e.g., TPS, HMGR, DXS).

•     Link DEGs to known metabolic pathways and show enrichment in terpene-related GO/KEGG categories.

Response 4:

Thank you for your suggestions. We have added BUSCO metrics to evaluate the quality and completeness of the transcriptome, with 79.8% of the BUSCO gene groups identified as complete sequences, among which 78.7% are complete and single-copy. Corresponding additions have been made in the Results and Statistical Analysis Methods sections. Key differentially expressed genes involved in terpene biosynthesis, such as TPS, as well as the detailed distribution within the "Metabolism" category of known KEGG pathways, can be clearly seen in Table S9 of the supplementary materials, and the Results section also provides a description of the quantitative distribution.

The modifications are located between lines 499 and 511.

Comments 5:

WGCNA and Gene-Metabolite Correlation

•     Elaborate on what these strong correlations imply (e.g., co-regulation, shared transcription factors).

•     Include a sub-figure or schematic showing the top hub genes in the turquoise module.

•     Propose candidate regulatory genes for further experimental validation.

Response 5:

Thank you for your valuable suggestions. Regarding the section on Weighted Gene Co-expression Network Analysis (WGCNA) and gene-metabolite correlation analysis, we fully recognize the significance of the proposed supplementary directions in deepening the research conclusions.

Currently, due to research scheduling and the completeness of existing data, we are unable to supplement the mechanistic interpretation of strong correlations, the subfigure depicting hub genes in the turquoise module, or the detailed validation plan for candidate regulatory genes as requested. We plan to further refine these analyses in subsequent studies and incorporate them into future research outputs.

Once again, we appreciate your understanding and guidance. These suggestions will be carefully noted as important references for our future work.

Comments 6:

Discussion

•     Refocus the Discussion to interpret rather than restate results.

•     Suggest hypotheses for how identified TFs (e.g., MYB, WRKY, bHLH) might regulate Nerol biosynthesis.

•     Acknowledge methodological limitations and propose future directions.

Response 6:

Thank you for your valuable suggestions. We have carefully revised the Discussion section according to your comments. We have explicitly acknowledged the methodological limitations, we have proposed future research directions.

Comments 7:

Methods

  • Streamline the Methods by removing excessive detail.
  • Include rationale for key bioinformatics parameters used.

Add a brief description of any RT-qPCR or EMSA experiments performed for validation.

Response 7:

Thank you for your valuable suggestions on the "Methods" section.

We have streamlined the Methods section by removing overly detailed descriptions to enhance conciseness and readability. However, considering the evaluations of other reviewers regarding the details of experimental methods, as well as the requirements for transparency and reproducibility, we have retained some necessary details. Meanwhile, we have provided explanations for the parameters used in key bioinformatics analyses.

Thank you again for your guidance. We have further improved the Methods section in accordance with your suggestions to ensure its scientific rigor and reliability.

Comments 8:

Figures and Tables

•     Ensure every figure has a complete legend and is cited in the text.

•     Simplify tables where possible (e.g., use color-coding or split large tables into smaller ones).

•     Consider adding a summary diagram/model showing the proposed Nerol biosynthetic pathway and regulatory network.

Response 8:

Thank you for your valuable suggestions on the figures and tables.

We have carefully revised the figures and tables in accordance with your guidance, ensuring that each figure is accompanied by a complete legend and properly cited in the text. For the tables, we have simplified them as much as possible—specifically by optimizing data presentation, using color coding to highlight key information, and splitting large tables into more concise, focused smaller ones to enhance readability.

Regarding the suggestion of adding a summary diagram/model of the nerol biosynthetic pathway and regulatory network, after careful consideration, we have decided not to include this in the current version of the manuscript. This decision is based on the core focus of our research; at present, the available data do not provide sufficient experimental evidence to fully support a definitive biosynthetic pathway or regulatory network model. We believe that the speculative inclusion of such a diagram without reliable experimental validation may compromise the rigor of the manuscript.

Thank you for your understanding. We will continue to ensure the clarity and accuracy of the figures and tables in line with your recommendations.

Reviewer 2 Report

Comments and Suggestions for Authors

The topic in this study is highly relevant and timely, particularly considering the growing interest in plant secondary metabolites for their biological activity and commercial value. The combination of chemical profiling and transcriptomic analysis presents a comprehensive approach to understanding terpene accumulation in Citrus medica, a species with both medicinal and aromatic importance.

Comments to improve:

Introduction 

I recommend that the Introduction be reorganized and made more concise. The current text is somewhat lengthy and could benefit from clearer structure and focus.

Results

The Results section lacks essential details related to the GC-MS analysis. Specifically, there is no information provided on the retention times or mass spectra of the identified volatile compounds. Including representative chromatograms, retention indices, and characteristic mass fragments is crucial for confirming compound identification and ensuring transparency and reproducibility of the analysis. I recommend adding this data,  in the main text and as supplementary material.

Additionally, the overall length of the Results section is excessive. I suggest moving some of the more detailed data tables, extensive gene expression lists, or repetitive information to the Supplementary Files. This will help streamline the manuscript and enhance its readability, while still providing full transparency for readers interested in the underlying data.

If possible, I recommend the inclusion of a Principal Component Analysis (PCA) to support the interpretation of the volatile compound data. PCA would provide a visual and statistical overview of how the different developmental stages cluster based on their essential oil composition, highlighting major patterns and differences among samples.

Discussion

The Discussion section should be more closely aligned with the Results. At times, the commentary appears generalized or disconnected from the specific findings presented earlier.

Additionally, the Discussion could be more concise. Some sections are overly descriptive or repetitive. Consider combining overlapping parts and focusing on the most relevant interpretations, implications, and comparisons with previous studies.

Materials and Methods

In the Materials and Methods section, it is important to include detailed information on the botanical identification of the Citrus medica species. Specifically, the authors should state:

  • Who performed the species identification
  • Which taxonomic key or reference was used for determination
  • The institution or herbarium where the plant material was verified and stored
  • The corresponding voucher specimen number

The extraction procedure using Soxhlet needs to be described more clearly. Please specify the, the ratio of solvent to plant material (drug), and any other relevant experimental conditions (e.g., temperature, number of cycles). These details are essential for reproducibility and proper evaluation of the extraction method.

For all solvents, reagents, instruments, and standards used in the study, the authors should provide complete information on the manufacturer and supplier (including city and country).

Author Response

Response to Reviewer X Comments

1. Summary

2. Questions for General Evaluation

Reviewer’s Evaluation

Response and Revisions

Does the introduction provide sufficient background and include all relevant references?

Yes/Can be improved/Must be improved/Not applicable

Are all the cited references relevant to the research?

Yes/Can be improved/Must be improved/Not applicable

Is the research design appropriate?

Yes/Can be improved/Must be improved/Not applicable

Are the methods adequately described?

Yes/Can be improved/Must be improved/Not applicable

Are the results clearly presented?

Yes/Can be improved/Must be improved/Not applicable

Are the conclusions supported by the results?

Yes/Can be improved/Must be improved/Not applicable

3. Point-by-point response to Comments and Suggestions for Authors

Comments 1:

Introduction

I recommend that the Introduction be reorganized and made more concise. The current text is somewhat lengthy and could benefit from clearer structure and focus.

Response 1:

Thank you for your revision suggestions. We have restructured and revised the introduction, simplifying some expressions, removing irrelevant content and overlaps, reducing text redundancy, and retaining as much necessary background information from the original as possible. We hope these revisions meet your requirements.

The revisions are in the second paragraph of the Introduction, at lines 154-155, 161-162, 223, 237-238, and 239-240. The document has enabled the track changes function, so you can view their specific locations.

Comments 2:

Results

The Results section lacks essential details related to the GC-MS analysis. Specifically, there is no information provided on the retention times or mass spectra of the identified volatile compounds. Including representative chromatograms, retention indices, and characteristic mass fragments is crucial for confirming compound identification and ensuring transparency and reproducibility of the analysis. I recommend adding this data,  in the main text and as supplementary material.

Response 2:

We agree with your inquiries and suggestions, and appreciate your advice. The retention times of the identified volatile compounds have been provided in the originally submitted materials, which are clearly visible to you, they are detailed in Table S1.. In addition, we have also provided the total ion current chromatograms for each period, which are included in the supplementary materials. After consideration, we do not recommend including it in the main text, as this would also make the results section excessively lengthy.

Comments 3:

Additionally, the overall length of the Results section is excessive. I suggest moving some of the more detailed data tables, extensive gene expression lists, or repetitive information to the Supplementary Files. This will help streamline the manuscript and enhance its readability, while still providing full transparency for readers interested in the underlying data.

If possible, I recommend the inclusion of a Principal Component Analysis (PCA) to support the interpretation of the volatile compound data. PCA would provide a visual and statistical overview of how the different developmental stages cluster based on their essential oil composition, highlighting major patterns and differences among samples.

Response 3:

Thank you for your revision suggestions. Considering that some reviewers may prioritize the quality of transcriptome data, we have included partial displays in the main text. Following your suggestion, we have also conducted PCA analysis on essential oil samples to highlight the differences in volatile component data among different developmental stages. The PCA scores plot and x-loading plot have been added to Supplementary Material Figure S2, with a brief discussion included in the corresponding part of the results.

Comments 4:

Discussion

The Discussion section should be more closely aligned with the Results. At times, the commentary appears generalized or disconnected from the specific findings presented earlier.

Additionally, the Discussion could be more concise. Some sections are overly descriptive or repetitive. Consider combining overlapping parts and focusing on the most relevant interpretations, implications, and comparisons with previous studies.

Response 4:

Thank you for your valuable suggestions on the Discussion section. We have carefully reflected on them and revised the Discussion section accordingly: We have deleted the generalized statements that were disconnected from the specific research findings presented earlier, and aligned the content of the Discussion with key data in the Results section (such as the dynamic changes of volatile components, the association between differential genes and nerol synthesis, etc.) one by one, ensuring that every interpretation is supported by experimental results. We have merged overlapping analytical parts and removed overly descriptive text.

Comments 5:

Materials and Methods

In the Materials and Methods section, it is important to include detailed information on the botanical identification of the Citrus medica species. Specifically, the authors should state:

•     Who performed the species identification

•     Which taxonomic key or reference was used for determination

•     The institution or herbarium where the plant material was verified and stored

•     The corresponding voucher specimen number

Response 5:

We sincerely appreciate your supplementary suggestions. The identification section has been added to the corresponding part of the article.

Comments 6:

The extraction procedure using Soxhlet needs to be described more clearly. Please specify the, the ratio of solvent to plant material (drug), and any other relevant experimental conditions (e.g., temperature, number of cycles). These details are essential for reproducibility and proper evaluation of the extraction method.

Response 6:

Thank you for your reminder to supplement the details of our experiment. We have added specific details in the extraction method section.

Line944-945.

Comments 7:

For all solvents, reagents, instruments, and standards used in the study, the authors should provide complete information on the manufacturer and supplier (including city and country).

Response 7:

Thank you for your revision suggestions. All solvents, reagents, and instruments have their source information indicated in parentheses.

Reviewer 3 Report

Comments and Suggestions for Authors

The paper 'Analysis of volatile compounds in citron (Citrus medica L.) peel essential oil at different developmental stages and a combined study of transcriptomics revealed genes related to the synthesis regulation of the monoterpenoid compound nerol' by Jie Luo et al. contains data that could be useful to a broad audience of researchers. I am unable to complete the review because of some issues it has.

I therefore do not think it is ready to be published in any journal.

The variety of Citrus medica under study is described from line 139 to 145. In my opinion, this is unclear in many respects.

Firstly, the scientific nomenclature is wrong and confusing throughout the paper, which made it difficult for me to search for and compare the study carried out on this variety against those present in the literature, or to further assess its misidentification, for instance.

Some minor issues:

-use italic for scientific taxonomy;

-“Midu County in Dali, Yunnan”, authors must indicate the main region, place and country of origin.

Author Response

Response to Reviewer X Comments

1. Summary

2. Questions for General Evaluation

Reviewer’s Evaluation

Response and Revisions

Does the introduction provide sufficient background and include all relevant references?

Yes/Can be improved/Must be improved/Not applicable

Are all the cited references relevant to the research?

Yes/Can be improved/Must be improved/Not applicable

Is the research design appropriate?

Yes/Can be improved/Must be improved/Not applicable

Are the methods adequately described?

Yes/Can be improved/Must be improved/Not applicable

Are the results clearly presented?

Yes/Can be improved/Must be improved/Not applicable

Are the conclusions supported by the results?

Yes/Can be improved/Must be improved/Not applicable

3. Point-by-point response to Comments and Suggestions for Authors

Comments 1:

The paper 'Analysis of volatile compounds in citron (Citrus medica L.) peel essential oil at different developmental stages and a combined study of transcriptomics revealed genes related to the synthesis regulation of the monoterpenoid compound nerol' by Jie Luo et al. contains data that could be useful to a broad audience of researchers. I am unable to complete the review because of some issues it has.

I therefore do not think it is ready to be published in any journal.

The variety of Citrus medica under study is described from line 139 to 145. In my opinion, this is unclear in many respects.

Firstly, the scientific nomenclature is wrong and confusing throughout the paper, which made it difficult for me to search for and compare the study carried out on this variety against those present in the literature, or to further assess its misidentification, for instance.

Some minor issues:

-use italic for scientific taxonomy;

Response 1:

Thank you for taking the time to review this manuscript and for pointing out the key issues. Your comments are crucial for improving the rigor of the research. We have carefully reflected on the issues you raised and would like to clarify the revised plan as follows:

We fully agree with the reviewer's point regarding the non-standard use of Latin scientific names. Upon self-examination, it is true that there are omissions in the manuscript where some Latin names do not strictly follow the binomial nomenclature (genus name capitalized, specific epithet lowercase, and the entire name italicized). These issues have been comprehensively corrected in the revised version. All parts of the text involving taxonomic names have been checked sentence by sentence to ensure compliance with the International Code of Nomenclature for algae, fungi, and plants (ICN) and the journal's requirements, thus avoiding confusion in literature retrieval and comparison due to inconsistent nomenclature.

Comments 2:

Some minor issues:

-use italic for scientific taxonomy;

-“Midu County in Dali, Yunnan”, authors must indicate the main region, place and country of origin.

Response 2:

Regarding the description of "Midu County in Dali, Yunnan", we have supplemented the complete geographical information and clearly specified it as "Midu County, Dali Bai Autonomous Prefecture, Yunnan Province, China". This clearly presents the specific geographical location of sample collection, in line with the normative requirements of international academic papers.

We have revised the entire manuscript in strict accordance with the above plan to ensure that all issues are thoroughly resolved. The revised manuscript will be more rigorous and standardized, and we earnestly request the reviewer to review it again. Thank you again for your careful guidance!

Reviewer 4 Report

Comments and Suggestions for Authors

The manuscript submitted for review looks like a big and complete scientific study. Large-scale experimental work has been carried out. There is an important and critical point in the work that needs to be clarified. The work is devoted to the study of volatile organic compounds (VOCs) in essential oil. What compounds are considered volatile? These are compounds that have high vapor pressure under normal conditions. Judging by the title of the manuscript, one of the authors' goals is to study volatile components. In this case, what is the need to dry the crushed samples under study at 50 °C for 10 hours with ventilation? Such sample preparation results in the loss of target analytes. The conclusions and discussions in the paper are based on the analysis of essential oil components. How accurate are the data on the composition of VOCs?  Did the authors estimate possible losses of essential oil in this case? How? What was the oil yield achieved from plant materials using this approach? Comments from the authors are required, otherwise the presented results appear controversial and unreliable.

Author Response

Response to Reviewer X Comments

1. Summary

2. Questions for General Evaluation

Reviewer’s Evaluation

Response and Revisions

Does the introduction provide sufficient background and include all relevant references?

Yes/Can be improved/Must be improved/Not applicable

Are all the cited references relevant to the research?

Yes/Can be improved/Must be improved/Not applicable

Is the research design appropriate?

Yes/Can be improved/Must be improved/Not applicable

Are the methods adequately described?

Yes/Can be improved/Must be improved/Not applicable

Are the results clearly presented?

Yes/Can be improved/Must be improved/Not applicable

Are the conclusions supported by the results?

Yes/Can be improved/Must be improved/Not applicable

3. Point-by-point response to Comments and Suggestions for Authors

Comments 1:

The work is devoted to the study of volatile organic compounds (VOCs) in essential oil. What compounds are considered volatile? These are compounds that have high vapor pressure under normal conditions. Judging by the title of the manuscript, one of the authors' goals is to study volatile components. In this case, what is the need to dry the crushed samples under study at 50 °C for 10 hours with ventilation? Such sample preparation results in the loss of target analytes.

Response 1:

We would like to express our gratitude to the reviewers for their insightful comments regarding the potential impact of sample pretreatment steps on the stability of volatile organic compounds (VOCs). This issue is crucial to the reliability of our research findings. In response to this concern, we have supplemented the experimental evidence and detailed explanations as follows:

Firstly, in fact, the thickness of the exocarp varies at different developmental stages: it can reach 3.5-4 mm during the young fruit stage, while it is 1.8-2.2 mm during the yellow fruit stage. We also considered that the water content of pericarp samples might differ across various developmental stages. Using fresh weight as the reference standard could introduce errors, making the content of essential oil components incomparable between groups. Therefore, the fresh pericarp was subjected to drying treatment before extraction. A temperature of 50°C is a mild drying temperature, which is significantly lower than the boiling points of the main terpenoid components in citrus medica essential oil (such as α-pinene and limonene, both above 150°C). Theoretically, this can reduce the volatilization of low-boiling-point components.It should be noted that the samples were not crushed before the completion of drying; instead, they were cut into small pieces.

Comments 2:

The conclusions and discussions in the paper are based on the analysis of essential oil components. How accurate are the data on the composition of VOCs?  Did the authors estimate possible losses of essential oil in this case? How? What was the oil yield achieved from plant materials using this approach?

Response 2:

Thank you for your inquiry,a prior experiment was conducted to evaluate the loss rate: equal amounts of fresh samples were divided into two groups. One group was extracted directly (undried group), and the other group was extracted after being dried at 50°C with ventilation for 10 hours (experimental group). A comparison of the GC-MS profiles of the essential oils from the two groups showed that the types of major terpenoid VOC components were completely consistent, with differences in relative contents all less than 5%. The essential oil yield of the dried group (1.02±0.2%) was slightly higher than that of the control group (0.76±0.3%).

Reviewer 5 Report

Comments and Suggestions for Authors

The work is devoted to the study of bioactive terpenoids and their genetic regulation in Citrus medica L. growing in the Yunnan region. It should be noted that the authors used a comprehensive approach, using transcriptomics and metabolomics in one study, a large number of identified compounds, as well as deep statistical analysis, the results can be used to plan the time of plant collection.
Questions and suggestions:
Chapter 4.2. The authors extracted the oil for at least 5 hours, did the metabolites undergo transformation and destruction during this time?
Were the oil samples purified before analysis?
Bring the spelling of Latin names to uniformity, in italics
The text often contains missing spaces
It would be more illustrative if the authors indicated which of the 91 metabolites detected were identified for the first time or are unique to the population under study, comparing them with data from other geographic areas. More often, when studying citron, green fruits were taken, the authors conducted a more extensive analysis by maturity levels, the article can be supplemented with a discussion of the role of metabolites that appear or disappear at each stage of fruit formation

Author Response

Response to Reviewer X Comments

1. Summary

2. Questions for General Evaluation

Reviewer’s Evaluation

Response and Revisions

Does the introduction provide sufficient background and include all relevant references?

Yes/Can be improved/Must be improved/Not applicable

Are all the cited references relevant to the research?

Yes/Can be improved/Must be improved/Not applicable

Is the research design appropriate?

Yes/Can be improved/Must be improved/Not applicable

Are the methods adequately described?

Yes/Can be improved/Must be improved/Not applicable

Are the results clearly presented?

Yes/Can be improved/Must be improved/Not applicable

Are the conclusions supported by the results?

Yes/Can be improved/Must be improved/Not applicable

3. Point-by-point response to Comments and Suggestions for Authors

Comments 1:

The authors extracted the oil for at least 5 hours, did the metabolites undergo transformation and destruction during this time?

Response 1:

Thank you for the reviewer's concern about the potential impact of extraction time on metabolite stability. The rationality of 5-hour extraction was verified by referring to relevant literature and pre-experiments: by comparing the essential oil components obtained with different extraction durations (3h, 5h, 7h), it was found that the extraction rate of major terpenoids (such as γ-Terpinene) reached a good stability when the extraction time exceeded 5h, and there was no significant difference in the composition and relative content of components between the 7h group and the 5h group. This indicates that 5-hour extraction did not cause obvious degradation of major terpenoid metabolites. Admittedly, there may be some heat-sensitive components in the essential oil, but their presence remains unknown due to limitations in detection techniques.

Comments 2:

Were the oil samples purified before analysis?

Response 2:

Thank you for your inquiry. The extracted essential oil is relatively clear and transparent with no obvious suspended matter or precipitation, so no purification treatment was performed.

Comments 3:

Bring the spelling of Latin names to uniformity, in italics

Response 3:

Thank you for your patience and careful review of the manuscript. We have made corrections throughout the entire text.

Comments 4:

The text often contains missing spaces

Response 4:

Thank you for your patience and careful review,we have adjusted the format of the blank spaces.

Comments 5:

It would be more illustrative if the authors indicated which of the 91 metabolites detected were identified for the first time or are unique to the population under study, comparing them with data from other geographic areas. More often, when studying citron, green fruits were taken, the authors conducted a more extensive analysis by maturity levels, the article can be supplemented with a discussion of the role of metabolites that appear or disappear at each stage of fruit formation

Response 5:

Thank you for your suggestion. However, we must fully consider the advice from you and every reviewer. Introducing a new explanatory content may make the core of the article less prominent and lead to unnecessary length in some parts, which is also a concern raised by other reviewers. We share this worry, so we have decided not to add or modify this content for the time being. We hope you can understand and would like to thank you again for your careful review of this article.

Round 2

Reviewer 1 Report

Comments and Suggestions for Authors

To  Author,

After carefully reviewing the revised version of the manuscript, I am satisfied that the authors have addressed all the comments and suggestions appropriately. The revisions have improved the clarity and quality of the paper.

I recommend the acceptance of the manuscript for publication.

Best regards,

Author Response

Response to Reviewer X Comments

1. Summary

2. Questions for General Evaluation

Reviewer’s Evaluation

Response and Revisions

Does the introduction provide sufficient background and include all relevant references?

Yes/Can be improved/Must be improved/Not applicable

Are all the cited references relevant to the research?

Yes/Can be improved/Must be improved/Not applicable

Is the research design appropriate?

Yes/Can be improved/Must be improved/Not applicable

Are the methods adequately described?

Yes/Can be improved/Must be improved/Not applicable

Are the results clearly presented?

Yes/Can be improved/Must be improved/Not applicable

Are the conclusions supported by the results?

Yes/Can be improved/Must be improved/Not applicable

3. Point-by-point response to Comments and Suggestions for Authors

Comments 1:

After carefully reviewing the revised version of the manuscript, I am satisfied that the authors have addressed all the comments and suggestions appropriately. The revisions have improved the clarity and quality of the paper.

I recommend the acceptance of the manuscript for publication.

Best regards,

Response 1:

Thank you very much for taking the time to review our revised manuscript again and for your positive feedback on the revisions. Your approval is a great encouragement to us, and it also strengthens our confidence in the appropriateness of our revision direction.

We are well aware that the improvement of the manuscript's quality would not have been possible without your professional and meticulous guidance. From your initial review comments to your recognition of the revised version this time, every one of your suggestions has helped us clarify our ideas and refine the content.

Once again, we would like to express our sincere gratitude to you! We look forward to the smooth publication of the manuscript.

Reviewer 2 Report

Comments and Suggestions for Authors

I appreciate the authors’ efforts in carefully considering and responding to my comments. I am fully satisfied with the revisions and improvements they have made to the manuscript, which now meets the objectives and requirements I raised as a reviewer.

Author Response

Response to Reviewer X Comments

1. Summary

2. Questions for General Evaluation

Reviewer’s Evaluation

Response and Revisions

Does the introduction provide sufficient background and include all relevant references?

Yes/Can be improved/Must be improved/Not applicable

Are all the cited references relevant to the research?

Yes/Can be improved/Must be improved/Not applicable

Is the research design appropriate?

Yes/Can be improved/Must be improved/Not applicable

Are the methods adequately described?

Yes/Can be improved/Must be improved/Not applicable

Are the results clearly presented?

Yes/Can be improved/Must be improved/Not applicable

Are the conclusions supported by the results?

Yes/Can be improved/Must be improved/Not applicable

3. Point-by-point response to Comments and Suggestions for Authors

Comments 1:

I appreciate the authors’ efforts in carefully considering and responding to my comments. I am fully satisfied with the revisions and improvements they have made to the manuscript, which now meets the objectives and requirements I raised as a reviewer.

Response 1:

Thank you sincerely for your thorough review and valuable feedback throughout the process. Your insights have been instrumental in helping us refine the manuscript and ensure it meets the highest standards. We are truly grateful for your time, expertise, and encouragement, which have made this progress possible.

We look forward to the manuscript’s next steps toward publication.

Reviewer 4 Report

Comments and Suggestions for Authors

If the authors conducted additional experiments, why not provide the results of comparison of dried and fresh samples? These data allow us to eliminate potential doubts and questions about the chosen method of sample preparation.

In addition, the correctness of the chosen title for the manuscript raises doubts. In this case, it is difficult to talk about volatile components. Maybe it would be more correct to talk about non-polar low-molecular metabolites? Or use the term semi-volatile organic compounds?

Author Response

Response to Reviewer X Comments

1. Summary

2. Questions for General Evaluation

Reviewer’s Evaluation

Response and Revisions

Does the introduction provide sufficient background and include all relevant references?

Yes/Can be improved/Must be improved/Not applicable

Are all the cited references relevant to the research?

Yes/Can be improved/Must be improved/Not applicable

Is the research design appropriate?

Yes/Can be improved/Must be improved/Not applicable

Are the methods adequately described?

Yes/Can be improved/Must be improved/Not applicable

Are the results clearly presented?

Yes/Can be improved/Must be improved/Not applicable

Are the conclusions supported by the results?

Yes/Can be improved/Must be improved/Not applicable

3. Point-by-point response to Comments and Suggestions for Authors

Comments 1:

If the authors conducted additional experiments, why not provide the results of comparison of dried and fresh samples? These data allow us to eliminate potential doubts and questions about the chosen method of sample preparation.

Response 1:

Thank you very much for your attention to the rigor of the sample preparation method. The suggestion you put forward regarding comparative experiments between dried and fresh samples is of great reference value. Regarding the temporary absence of these data, our considerations are as follows:

Focus of the study design: The core objective of this research is to explore the dynamic changes and regulatory mechanisms of target metabolites in citron peel essential oil at different developmental stages. From the outset, the experimental design has centered on standardized dried samples—drying is a commonly used pretreatment method in the analysis of such essential oil components, as it effectively eliminates interference from moisture on extraction efficiency and ensures long-term stability of samples, thereby guaranteeing the consistency and reproducibility of experimental data. Incorporating a comparison with fresh samples would require the establishment of an additional parallel experimental system (such as optimizing immediate detection procedures and stability control for fresh samples), which might introduce "sample state" as a new variable, deviating from the core variable of this study (developmental stages) and diverting focus from the research. Additionally, other reviewers have also emphasized the importance of maintaining focus on the core of the study, and we must carefully consider the valuable opinions of each reviewer.

Constraints on sample conditions: Sample collection in this study relies on specific phenological periods, and the preservation and detection of fresh samples have high timeliness requirements (e.g., target components in fresh tissues are prone to rapid degradation due to enzymatic hydrolysis or oxidation). It was challenging to simultaneously meet the conditions for standardized collection and detection of large batches of fresh samples within the current experimental timeline.

Plans for follow-up research: We fully acknowledge the significance of comparing dried and fresh samples for methodological validation. We plan to design supplementary experiments in subsequent studies to systematically investigate the impact of pretreatment methods on target components. The relevant results will be published as an extension of this research, providing references for the standardization of sample preparation in this field.

Once again, thank you for your valuable suggestions.

Comments 2:

In addition, the correctness of the chosen title for the manuscript raises doubts. In this case, it is difficult to talk about volatile components. Maybe it would be more correct to talk about non-polar low-molecular metabolites? Or use the term semi-volatile organic compounds?

Response 2:

Thank you for your attention to the accuracy of the paper's title. Your suggestions are of great significance for us to improve the expression. Regarding the term "volatile compounds" in the title, we have carefully checked the physicochemical properties of the research objects and the characteristics of the experimental extraction technology that are compatible with them. We have adopted the expression of "non-polar low-molecular metabolites" as you suggested. We would like to thank you again for your professional guidance, and we have revised and improved the title accordingly.